# Dynamic chromatin architecture identifies new autoimmune-associated enhancers for *IL2* and novel genes regulating CD4+ T cell activation

Matthew C Pahl[1,2†], Prabhat Sharma[1,3†], Rajan M Thomas[1,3†], Zachary Thompson[1,3], Zachary Mount[1,3], James A Pippin[1,2], Peter A Morawski[4], Peng Sun[1,3], Chun Su[1,2], Daniel Campbell[4,5], Struan FA Grant[1,2,6,7,8], Andrew D Wells[1,3,9,10]*

[1]Center for Spatial and Functional Genomics, Children's Hospital of Philadelphia, Philadelphia, United States; [2]Division of Human Genetics, The Children's Hospital of Philadelphia, Philadelphia, United States; [3]Department of Pathology, The Children's Hospital of Philadelphia, Philadelphia, United States; [4]Benaroya Research Institute at Virginia Mason, Seattle, United States; [5]Department of Immunology, University of Washington School of Medicine, Seattle, United States; [6]Department of Genetics, Perelman School of Medicine, University of Pennsylvania, Philadelphia, United States; [7]Department of Pediatrics, Perelman School of Medicine, University of Pennsylvania, Philadelphia, United States; [8]Division of Endocrinology and Diabetes, The Children's Hospital of Philadelphia, Philadelphia, United States; [9]Department of Pathology and Laboratory Medicine, Perelman School of Medicine, University of Pennsylvania, Philadelphia, United States; [10]Institute for Immunology, Perelman School of Medicine, University of Pennsylvania, Philadelphia, United States

*For correspondence:
adwells@pennmedicine.upenn.edu

†These authors contributed equally to this work

Competing interest: The authors declare that no competing interests exist.

**Abstract** Genome-wide association studies (GWAS) have identified hundreds of genetic signals associated with autoimmune disease. The majority of these signals are located in non-coding regions and likely impact *cis*-regulatory elements (cRE). Because cRE function is dynamic across cell types and states, profiling the epigenetic status of cRE across physiological processes is necessary to characterize the molecular mechanisms by which autoimmune variants contribute to disease risk. We localized risk variants from 15 autoimmune GWAS to cRE active during TCR-CD28 co-stimulation of naïve human CD4+ T cells. To characterize how dynamic changes in gene expression correlate with cRE activity, we measured transcript levels, chromatin accessibility, and promoter–cRE contacts across three phases of naive CD4+ T cell activation using RNA-seq, ATAC-seq, and HiC. We identified ~1200 protein-coding genes physically connected to accessible disease-associated variants at 423 GWAS signals, at least one-third of which are dynamically regulated by activation. From these maps, we functionally validated a novel stretch of evolutionarily conserved intergenic enhancers whose activity is required for activation-induced *IL2* gene expression in human and mouse, and is influenced by autoimmune-associated genetic variation. The set of genes implicated by this approach are enriched for genes controlling CD4+ T cell function and genes involved in human inborn errors of immunity, and we pharmacologically validated eight implicated genes as novel regulators of T cell activation. These studies directly show how autoimmune variants and the genes they regulate influence processes involved in CD4+ T cell proliferation and activation.

### eLife assessment

This is a **solid** study that follows a well-established canvas for variant-to-gene prioritization using 3D genomics, applying it to activated T cells. The authors go some way in validating the lists of candidate genes, as well as exploring the regulatory architecture of a candidate GWAS locus. Jointly with data from previous studies performing variant-to-gene assignment in activated CD4 T cells (and other immune cells), this work provides a **useful** additional resource for interpreting autoimmune disease-associated genetic variation.

## Introduction

Genome-wide association studies (GWAS) has linked hundreds of regions of the human genome to autoimmune disease susceptibility. The majority of GWAS variants are located in non-coding regions of the genome and likely contribute to disease risk by modulating *cis*-regulatory element (cRE) activity to influence gene expression (*Zhang and Lupski, 2015*). Identifying causal variants and effector genes molecularly responsible for increased disease risk is critical for identifying targets for downstream molecular study and therapeutic intervention. An understanding of how non-coding variants function is often limited by incomplete knowledge of the mechanism of action, that is, whether a variant is located in a cRE, in which cell types a cRE is active, and which genes are regulated by which cRE.

CD4+ T cells are key regulators of innate and adaptive immune responses that combat infection by orchestrating the activity of other immune cells. In their quiescent, naïve state, 'helper' T cells traffic between the blood and secondary lymphoid tissues as part of an immunosurveillance program maintained by transcription factors (TF) such as KLF2, TOB, and FOXO (*Weinreich et al., 2009*; *Tzachanis et al., 2001*; *Yusuf and Fruman, 2003*). Upon encountering specific antigen, a cascade of signals activated through the TCR and co-stimulatory receptors results in large-scale changes in gene expression driven by factors like NFKB, NFAT, IRF4, and STATs, leading to proliferation and differentiation into specialized Th1, Th2, and Th17 subsets capable of participating in protective immunity (*Zhu et al., 2010*; *Saravia et al., 2019*). CD4+ T cells are also key players in the induction and pathogenesis of autoimmunity. The *cis*-regulatory architecture of CD4+ T cells is enriched for autoimmune disease GWAS variants (*Farh et al., 2015*; *Chen et al., 2016*; *Soskic et al., 2019*; *Robertson et al., 2021*; *Lu et al., 2021*; *Chandra et al., 2021*; *Soskic et al., 2022*; *Mouri et al., 2022*), and T cells from autoimmune patients harbor distinct epigenetic and transcriptomic signatures linking dysregulated gene expression with disease pathogenesis. Autoimmune variants may contribute to the breakdown of immune self-tolerance by shifting the balance between autoreactive conventional vs. regulatory CD4+ T cells, altering cytokine production, or promoting auto-antigen production (*Rioux and Abbas, 2005*; *Buckner, 2010*).

Chromatin conformation assays allow for identification of putative target genes of autoimmune variant-containing cRE in close spatial proximity to gene promoters. Previous work using promoter-capture HiC and HiC in combination with other epigenetic marks has implicated sets of autoimmune variants and effector genes that may participate in T cell activation (*Javierre et al., 2016*; *Burren et al., 2017*; *Yang et al., 2020*). In addition to chromatin conformation approaches, multiple complementary approaches have been developed to link disease-associated SNPs to their downstream effector genes. Expression quantitative trait mapping (eQTL) and chromatin co-accessibility data can be used to implicate effector genes through statistical association of genotypes with readouts of expression or other molecular markers (*Bykova et al., 2022*). However, these approaches do not account for the relevant three-dimensional (3D) structure of the genome in the nucleus and are highly susceptible to *trans*-effects and other confounding factors.

In this study, we measured the impact of TCR-CD28 activation on the autoimmune-associated *cis*-regulatory architecture of CD4+ helper T cells, and by comparing our data to those of several orthogonal GWAS effector gene nomination studies, identify hundreds of effector genes not implicated previously. We find that 3D chromatin-based approaches exhibit two- to tenfold higher predictive sensitivity than eQTL and ABC statistical approaches when benchmarked against a 'gold standard set' of genes underlying inborn errors in immunity and tolerance. Our maps of autoimmune SNP-gene contacts also predicted a stretch of evolutionarily conserved, intergenic enhancers that we show are required for normal expression of the canonical T cell activation gene *IL2* in both human and mouse, whose activity is influenced by autoimmune risk variants. The set of variant-connected effector genes

defined by 3D physical proximity to autoimmune-associated cRE is enriched for genes that regulate T cell activation, as validated pharmacologically in this study and by CRISPR-based screens in orthogonal studies (*Shifrut et al., 2018*; *Schmidt et al., 2022*; *Freimer et al., 2022*).

## Results

### Gene expression dynamics as a function of naïve CD4+ T cell activation

To explore the universe of genes and cRE that are affected by and may contribute to CD4+ activation, we characterized the dynamics of gene expression, chromatin accessibility, and 3D chromatin conformation in human CD3+CD4+CD45RA +CD45RO-naïve T cells purified directly ex vivo and in response to in vitro activation through the T cell receptor (TCR) and CD28 for 8 or 24 hr using RNA-seq (N = 3 donors), ATAC-seq (N = 3 donors), and HiC (N = 2 donors, *Figure 1A*). As sequencing depth and restriction enzyme-cutting frequency affect the resolution and reliability of HiC experiments (*Su et al., 2021b*), we constructed libraries with two four-cutter restriction enzymes and sequenced to a total of approximately 4 billion unique-valid reads per timepoint. We verified the reproducibility of replicate ATAC-seq and RNA-seq libraries with principal component analysis and HiC libraries using distance-controlled, stratum-adjusted correlation coefficient (SCC). Replicate samples were highly correlated and clustered by activation stage (*Figure 1—figure supplement 1A–C*). As expected, quiescent naïve CD4+ T cells expressed high levels of *SELL, TCF7, CCR7,* and *IL7R,* and rapidly upregulated *CD69, CD44, HLADR,* and *IL2RA* upon stimulation (*Figure 1B*, *Supplementary file 1*). Genome-scale gene set variance analysis based on MsigDB hallmark pathways showed that genes involved in KRAS and Hedgehog signaling are actively enriched in quiescent naive CD4+ T cells, while stimulated cell transcriptomes are enriched for genes involved in cell cycle, metabolism, and TNF-, IL-2/STAT5-, IFNg-, Notch-, and MTORC1-mediated signaling pathways (*Supplementary file 2*).

To define global gene expression dynamics during the course of CD4+ T cell activation, we performed pairwise comparisons and k-means clustering, identifying 4390 differentially expressed genes after 8 hr of stimulation (3289 upregulated and 1101 downregulated) and 3611 differentially expressed genes between 8 hr and 24 hr (3015 upregulated and 596 downregulated, *Figure 1—figure supplement 1D*) that could be further separated into five clusters based on their distinct trajectories (*Figure 1C*, *Figure 1—figure supplement 1E–G*, *Supplementary file 1*). Genes upregulated early upon activation (cluster 1; n = 1621 genes) are enriched for pathways involved in the unfolded protein response, cytokine signaling, and translation (e.g., *IL2, IFNG, TNF, IL3, IL8, IL2RA, ICOS, CD40LG, FASLG, MYC, FOS, JUNB, REL, NFKB1, STAT5A, Supplementary file 3*). Genes downregulated late (cluster 2; n = 1600 genes) are moderately enriched for receptor tyrosine kinase signaling, cytokine signaling, and extracellular matrix organization. Genes monotonically increasing (cluster 3; n = 2676 genes) are highly enriched for pathways involved in infectious disease and RNA stability, translation and metabolism, and moderately enriched for pathways involved in the unfolded protein response, cellular responses to stress, regulation of apoptosis, and DNA repair (e.g., *TBX21, BHLHE40, IL12RB2, STAT1, CCND2, CDK4, PRMT1, ICAM1, EZH2, Supplementary file 3*). Genes downregulated early (cluster 4; n = 1628 genes) are enriched for pathways involved in inositol phosphate biosynthesis, neutrophil degranulation, and metabolism of nucleotides (e.g., *KLF2, IL7R, RORA, IL10RA*), and genes upregulated late (cluster 5; n = 2154 genes) are highly enriched for pathways involved in cell cycle, DNA unwinding, DNA repair, chromosomal maintenance, β-oxidation of octanoyl-CoA, and cellular response to stress (e.g., *CDK2, E2F1, CDK1, CCNE1, CCNA2, PCNA, WEE1, CDC6, ORC1, MCM2*). These patterns are consistent with known changes in the cellular processes that operate during T cell activation, confirming that in vitro stimulation of CD4+ T cells recapitulates gene expression programs known to be engaged during a T cell immune response.

### Dynamic changes in chromosomal architecture and genome accessibility during naïve CD4+ T cell activation

To understand the *cis*-regulatory dynamics underlying the observed activation-induced changes in gene expression, we examined CD4+ T cell nuclear chromosome conformation and chromatin accessibility as a function of stimulation state using HiC and ATAC-seq. The human genome consists of ~3 m of DNA that is incorporated into chromatin and compacted into the ~500 cubic micron nucleus of a cell in a hierarchically ordered manner. This degree of compaction results in only ~1% of genomic

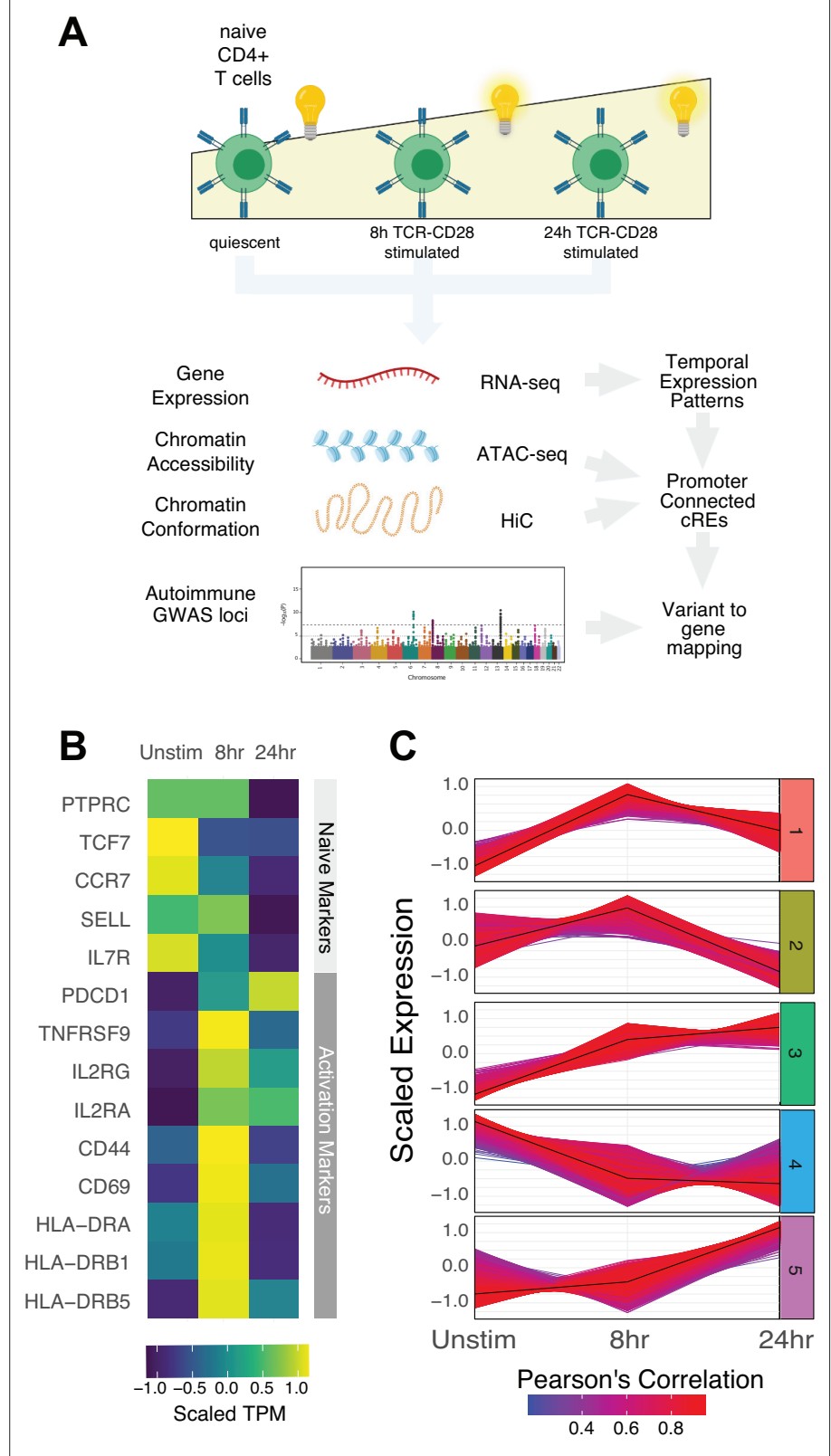

**Figure 1.** Defined gene expression dynamics throughout activation of naïve CD4+ T cells. (**A**) Diagram of study design: RNA-seq, ATAC-seq, and HiC libraries were prepared from sorted CD3+CD4+CD45RA+CD45RO- naïve T cells (donor N = 3) prior to or following 8 hr or 24 hr stimulation with anti-CD3/28 beads. *cis*-regulatory elements (cRE) are classified to genes allowing for interrogation of the pattern of expression, accessibility, and chromatin

*Figure 1 continued on next page*

*Figure 1 continued*

structure changes of autoimmune-associated variants. (**B**) Heatmap showing normalized expression of known markers of T cell activation. (**C**) k-means clustering of differentially expressed genes into five groups using the elbow and within-cluster-sum of squares methods to select the number of clusters. The centroid of the cluster is depicted as a black line, and members of the cluster are depicted as colored lines. Color indicates the Pearson's correlation coefficient between the gene and the cluster centroid, with red indicating higher correlation and blue lower correlation.

The online version of this article includes the following figure supplement(s) for figure 1:

**Figure supplement 1.** Library sequencing reproducibility and expression clustering.

DNA being accessible to the machinery that regulates gene transcription (*Thurman et al., 2012*); therefore, a map of open chromatin regions (OCR) in a cell represents its potential gene regulatory landscape. Open chromatin mapping of human CD4+ T cells at all states identified a consensus *cis*-regulatory landscape of 181,093 reproducible OCR (*Supplementary file 4*). Of these, 14% (25,291) exhibited differential accessibility following 8 hr of stimulation (false discovery rate[FDR] < 0.05). Most differentially accessible regions (DARs) became more open (18,887), but some DARs (6629) showed reduced accessibility (*Figure 2A*, *Supplementary file 5*). The change in accessibility over the next 16 hr of stimulation showed the opposite dynamic, with 6629 regions exhibiting reduced accessibility and only 4417 DARs becoming more open (total of 11,046 DARs, *Figure 2A*, *Supplementary file 5*). These OCR represent the set of putative cRE with dynamic activity during T cell activation.

The vast majority of putative cRE are located in intergenic or intronic regions of the genome far from gene promoters, meaning that the specific impact of a given cRE on gene expression cannot be properly interpreted from a one-dimensional (1D) map of genomic or epigenomic features. To predict which cRE may regulate which genes in CD4 +T cells across different states of activation, we created 3D maps of cRE-gene proximity in the context of genome structure. The highest order of 3D nuclear genome structure is represented by A/B compartments, which are large chromosomal domains that self-associate into transcriptionally active (A) vs. inactive (B) regions (*Lieberman-Aiden et al., 2009*). In agreement with prior studies, we find that OCR located in active A compartments exhibit higher average accessibility than those OCR located in less active B compartments (*Figure 2— figure supplement 1A*, *Supplementary file 6*), a trend observed across all cell states. Likewise, genes located in A compartments show higher average expression than those located in B compartments (*Figure 2—figure supplement 1B*). A quantitative comparison across cell states showed that 94% of the CD4+ T cell genome remained stably compartmentalized into A (42%) and B (52%), indicating that activation does not cause a major shift in the large-scale organization of the genome within the nucleus (*Figure 2B*).

Within each A or B compartment, the genome is further organized into topologically associating domains (TADs) (*Dixon et al., 2012*). These structures are defined by the fact that genomic regions within them have the potential to interact with each other in 3D, but have low potential to interact with regions outside the TAD. The location of TAD boundaries can influence gene expression by limiting the access of cRE to specific, topologically associated genes. While ~80% of TAD boundaries remained stable across all states, 20% of TAD boundaries (8925) changed as a function of T cell activation (*Figure 2C*, *Figure 2—figure supplement 1C and D*, *Supplementary file 7*). TAD boundary dynamics were categorized and 2198 boundaries exhibited a change in strength, 2030 boundaries shifted position, and 4697 boundaries exhibited more complex changes such as loss of a boundary resulting in merger of two neighboring TAD, addition of a boundary splitting one TAD into two, or a combination of any of these changes. Genes nearby dynamic TAD boundaries are enriched for pathways involved in RNA metabolism, cellular response to stress, and the activity of PTEN, p53, JAK/STAT, Runx, and Hedgehog (*Supplementary file 6*). We also detected chromatin stripes, which are TAD-like structures that consist of a genomic anchor region that is brought into contact with a larger loop domain via an active extrusion mechanism. Chromatin stripes are contained within and/or overlap TAD regions, and are enriched for active enhancers and super-enhancers (*Mirny et al., 2019*; *Yoon et al., 2022*). We identified 1526 chromatin stripes in quiescent naïve CD4+ T cells, 1676 stripes in 8 hr stimulated cells, and 2028 stripes at 24 hr post-stimulation (*Figure 2—figure supplement 2A*, *Supplementary file 8*). Consistent with prior studies in other cell types, chromatin stripes were

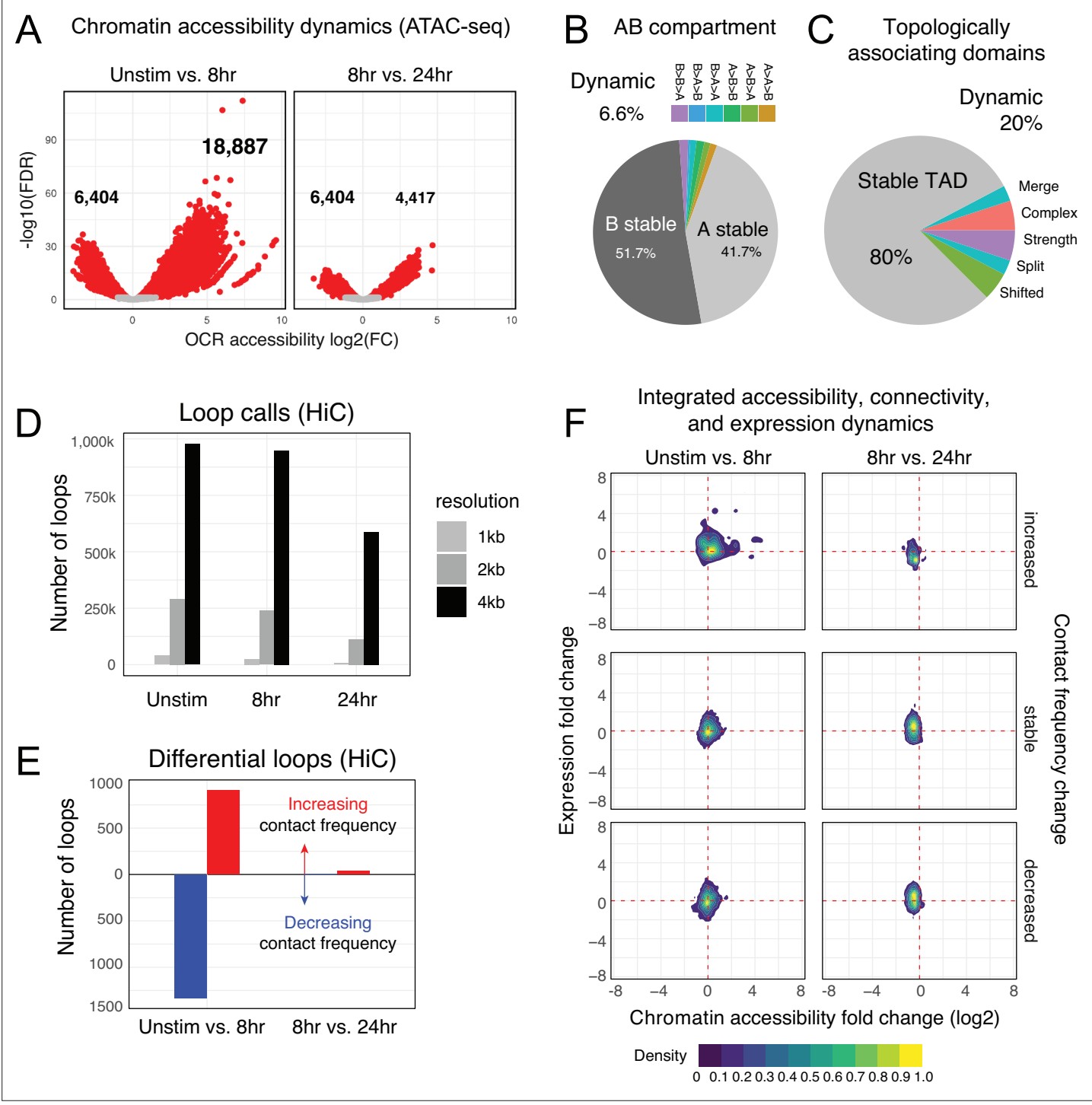

**Figure 2.** Chromatin accessibility and architecture changes across three stages of T cell activation. (**A**) Volcano plots depicting the log₂ change in accessibility in all reproducible open chromatin regions (OCR) (present in two of the three replicates) compared to the -log₁₀(FDR), significant points are indicated in red (FDR < 0.05; abs(log₂FC) > 1). (**B**) Genomic regions were classified by membership in A vs B compartments at each timepoint. The pie chart depicts regions with differential A/B compartment assignment in response to activation. (**C**) Topologically associating domain (TAD) structure was determined at each timepoint, and regions exhibiting a shift, split/merge, change in strength, or more complex rearrangements are depicted in the pie chart. (**D**) Loop calls identified from HiC data for each timepoint called at three resolutions (1 kb, 2 kb, 4 kb bins). (**E**) Differential loop calls called across all resolutions. (**F**) Density plots of *cis*-regulatory elements (cRE) accessibility and gene expression separated by whether contact frequency increased, decreased, or remained stable from the transition from unstimulated to 8 hr activation and 8 hr to 24 hr activation.

The online version of this article includes the following figure supplement(s) for figure 2:

*Figure 2 continued on next page*

**Figure supplement 1.** Characterization of CD4+ T cell epigenomic data.

**Figure supplement 2.** Characterization of chromatin stripes during CD4+ T cell activation.

**Figure supplement 3.** Large-scale epigenetic changes at dynamically expressed genes during CD4+ T cell activation.

preferentially located in A compartments, and genes and OCR within stripe regions showed increased expression and chromatin accessibility (*Figure 2—figure supplement 2B–D*).

Within active topological structures, transcriptional enhancers can regulate the expression of distant genes by looping to physically interact with gene promoters (*Kim and Shiekhattar, 2015*; *Hamamoto and Fukaya, 2022*). To identify potential regulatory cRE-gene interactions, we identified high-confidence loop contacts across all cell states using Fit-HiC (merged 1 kb, 2 kb, and 4 kb resolutions, *Figure 2D*). This approach detected 933,755 loop contacts in quiescent naïve CD4+ T cells, 900,267 loop contacts in 8 hr stimulated cells, and 551,802 contacts in 24 hr stimulated cells (2,099,627 total unique loops). Approximately 23% of these loops involved a gene promoter at one end and an OCR at the other, and these promoter-interacting OCR were enriched for enhancer signatures based on flanking histone marks from CD4+ T cells in the epigenome roadmap database (*Consortium, 2015*; *Figure 2—figure supplement 1E*). T cell activation resulted in significant reorganization of the open chromatin-promoter interactome as 907 promoter–OCR exhibited increased contact and 1333 showed decreased contact following 8 hr of stimulation (*Figure 2E*). Continued stimulation over the next 16 hr was associated with an increase in the contact frequency of 41 promoter–OCR pairs, while only 4 pairs exhibited decreased contact (*Figure 2E*). Activation-induced changes in chromatin architecture and gene expression were highly correlated, as genomic regions exhibiting increased promoter connectivity became more accessible at early stages of stimulation, which was associated with increased gene transcription from connected promoters (*Figure 2F*, *Supplementary file 9*). The accessibility of promoter-connected OCR and the expression of their associated genes decreased globally from 8 to 24 hr of stimulation (*Figure 2F*). We compared these loop calls to a prior chromatin capture analysis in CD4+ T cells by *Burren et al., 2017* and found that roughly 40% of stable loops in both stimulated and unstimulated cells were identical in both studies, despite differing in approach (HiC vs. PCHiC), analysis (HiC vs. CHiCAGO), sample (naïve CD4+ vs. total CD4+), timepoint (8 vs. 4 hr), and donor individuals (*Figure 2—figure supplement 1F*). As expected, unstimulated samples were more similar than activated samples.

We next focused on the five sets of genes with the dynamic expression patterns defined in *Figure 1D* and identified 57,609 OCR that contact dynamic gene promoters in at least one stage. Most dynamic genes contacted between 1 and ~35 OCR, with a median of 10 OCR per gene, but a handful of dynamic genes were observed in contact with over 100 distinct OCR (*Figure 2—figure supplement 1G*). Similarly, most OCR were connected to a single dynamic gene, but many contacted more than one gene (median 2 genes per OCR), suggesting that most dynamic genes have a complex regulatory architecture. Increased gene expression upon activation correlated with an increase in the accessibility and promoter contact frequency of distant cRE (*Figure 2—figure supplement 1H*), as exemplified by *GEM* and *IRF4* (*Figure 2—figure supplement 3A and B*). Conversely, the 3D regulatory architecture of genes like *KLF2* and *DPEP2*, which were downregulated following stimulation, exhibited decreased contact and accessibility (*Figure 2—figure supplement 3C and D*).

## Transcription factor footprints enriched in dynamic open chromatin identify regulators of T cell activation

To explore what factors drive dynamic changes in the regulatory architecture of the CD4+ T cell genome during activation, we conducted a quantitative footprinting analysis at 1,173,159 TF motifs located in the consensus CD4+ T cell open chromatin landscape using the average accessibility of the region surrounding each motif as a measure of regulatory activity (*Supplementary file 10*). Activation of naïve CD4+ T cells resulted in increased chromatin accessibility around bZIP motifs at tens of thousands of genomic regions by 8 hr post-stimulation (*Figure 3A and C*), which was associated with increased expression of Fos and Jun family members constituting the AP-1 complex, as well as the bZIP factors BATF, BACH1, and BACH2 (*Figure 3A*). The activity of NFE2 and SMAD was increased without increased expression (*Figure 3A and C*), likely due to post-translational regulation of these

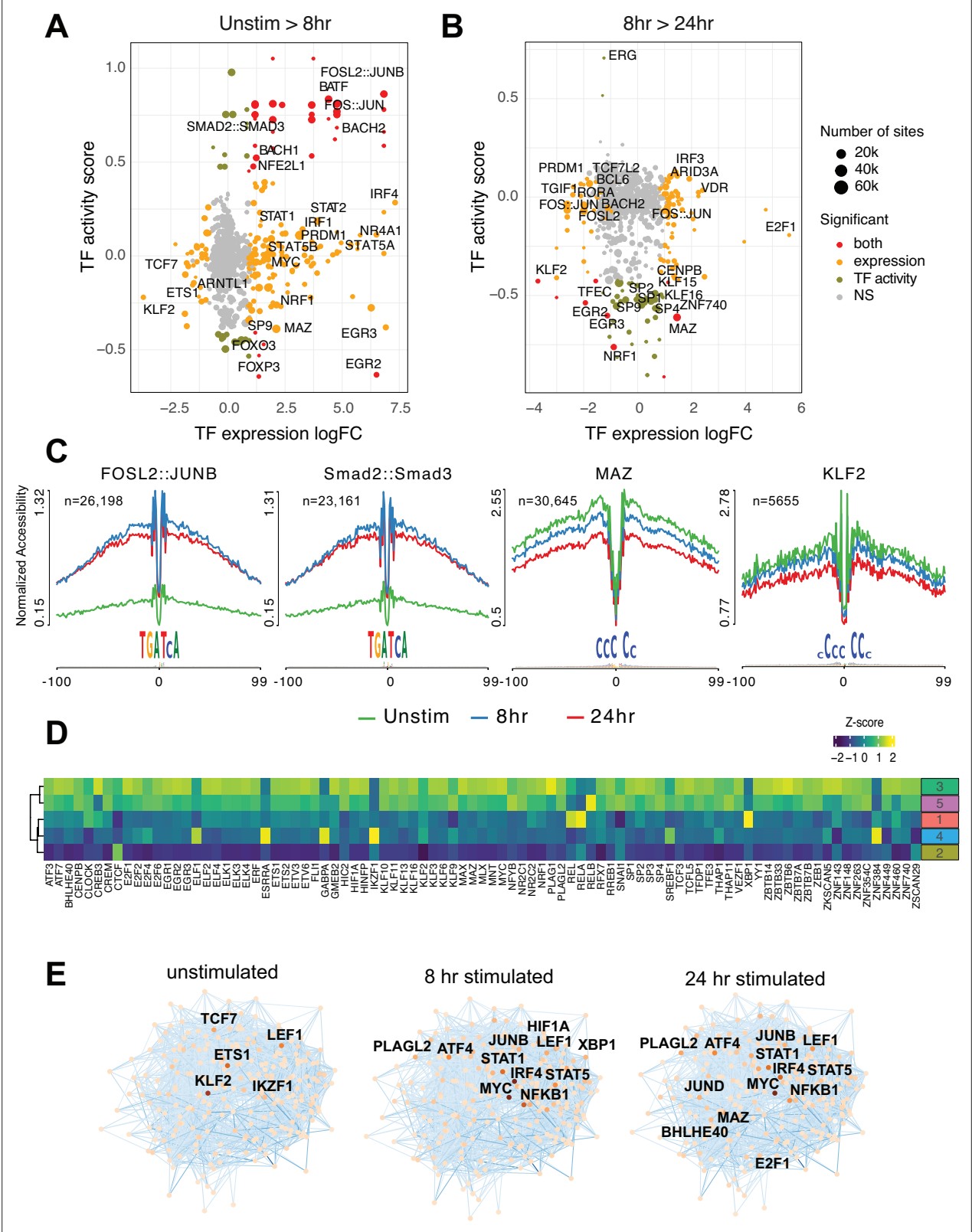

**Figure 3.** Transcription factors (TF) prediction potential regulators of chromatin status and expression changes. Footprints were annotated to each TF motif by sequence matching to the JASPAR motif database, and the average accessibility of the region surrounding each motif was used as a measure of activity at each timepoint. Scatterplots depict the change in accessibility for each TF motif (activity score) and log2FC of TF gene expression between unstimulated and 8 hr activation (**A**) or 8 hr and 24 hr activation (**B**). TF were classified as differentially expressed (orange), differentially active (brown),

*Figure 3 continued on next page*

*Figure 3 continued*

both (red), or neither (gray). Dot size indicates the number of predicted footprints occupied by each motif. (**C**) Average accessibility (normalized by depth and motif number) in a 200 bp window surrounding motif footprints for FOSL2::JUNB, Smad2::Smad3, MAZ, and KLF2 from three timepoints (unstim: green; 8 hr: blue; 24 hr: red). (**D**) Z-score of TF motif enrichment for *cis*-regulatory elements (cRE) connected to the five clusters compared to all open chromatin regions (OCR). Lighter color indicates higher specificity of enrichment for that TF cluster. (**E**) Connections between different TFs based on physical interactions and predicted regulation-based co-expression determined using GENIE3 for the three timepoints. Node color indicates expression (TPM; darker = higher expression, lighter = lower expression), edge color reflects confidence in the interaction called by GENIE3 (darker higher confidence).

The online version of this article includes the following figure supplement(s) for figure 3:

**Figure supplement 1.** Enrichment of transcription family members and gene regulatory network construction.

factors by phosphorylation (*Yoon et al., 2015*). Conversely, the motifs for a number of TF exhibited significantly reduced accessibility early after stimulation, including those for EGR2 and FOXP3 that are known to negatively regulate T cell activation (*Safford et al., 2005*; *Schubert et al., 2001*; *Figure 3A*). By 24 hr post-activation, bZIP activity remained largely unchanged compared to 8 hr (*Figure 3B*), but a number of factors showed decreased activity. These include several members of the Sp family, the Myc cofactor MAZ that also cooperates with CTCF to regulate chromatin looping (*Xiao et al., 2021*), KLF2, which controls genes involved in naïve CD4+ T cell quiescence and homing (*Weinreich et al., 2009*; *Sebzda et al., 2008*), NRF1, a factor implicated in age-associated T cell hypofunction (*Moskowitz et al., 2017*), and EGR2 and 3, which are known to oppose T cell activation and promote tolerance (*Safford et al., 2005*; *Harris et al., 2004*; *Williams et al., 2017*; *Figure 3B and C*).

To explore how TF activity may operate via the CD4+ T cell open chromatin landscape to regulate distinct programs of dynamic gene expression during TCR/CD28-induced activation, we focused on TF motifs enriched among those OCR specifically contacting promoters of dynamic genes identified by our clustering analysis (*Figure 3—figure supplement 1A*). The set of OCR contacting dynamic gene promoters were enriched for the motifs of 89 expressed (TPM > 5) TF compared to motifs present in the total open chromatin landscape (*Figure 3D*). The majority of this TF activity were enriched in OCR connected to genes highly upregulated at 24 hr post-activation (clusters 3 and 5, *Figure 3D*), with the exception of CREB3, ELF1, ESRRA, GABPA, RELA, XPB1, ZNF384, and the transcriptional repressor IKZF1 known as a strong negative regulator of T cell activation and differentiation (*Avitahl et al., 1999*; *Bandyopadhyay et al., 2007*; *Thomas et al., 2007*; *Thomas et al., 2010*). Conversely, motifs for IKZF1, ZNF384, GABPA, ESRRA, and ELF1 were highly enriched in the set of OCR contacting genes downregulated early after activation (cluster 4, *Figure 3D*). Motifs for KLF2 and the metabolic gene regulator SREBF1 were likewise enriched in OCR connected to downregulated genes. OCR interacting with genes in cluster 2 are negatively enriched for this set of TF except for CTCF (*Figure 3D*).

Finally, we integrated TF footprint, promoter connectome, and gene co-expression data to construct TF-gene regulatory networks likely operating at each timepoint. The connections between regulatory nodes are based on physical promoter–TF footprint interactions with confidence weighted by gene co-expression (*Figure 3E*, *Supplementary file 11*). Highly co-expressed genes at the core of the unstimulated CD4+ T cell regulatory network encode TF such as KLF2, ETS1, IKZF1, and TCF7 (*Weber et al., 2011*; *Johnson et al., 2018*; *Wang et al., 2022*) that are known to be involved in T cell gene silencing, quiescence, homeostasis, and homing. Genes connected to the top factors in this network were enriched for pathways involved in immune signaling, DNA replication and repair, protein secretion, and programmed cell death (*Figure 3—figure supplement 1B*). Co-stimulation through the TCR and CD28 induced a set of core network genes active at both timepoints with known roles in T cell activation and differentiation (*NFKB1, JUNB, MYC, IRF4, STAT5, STAT1, LEF1, ATF4*). Also part of this set is *PLAGL2*, an oncogene in the Wnt pathway that regulates hypoxia-induced genes (*Furukawa et al., 2001*) with no prior defined role in T cell activation. Additional nodes specifically implicated at 8 hr post-activation are HIF1A, the major sensor of cellular hypoxia (*Wu et al., 2022*), and XBP1, a major transcriptional mediator of the unfolded ER protein response with defined roles in T cell activation, differentiation, and exhaustion (*Fu et al., 2012*; *Pramanik et al., 2018*). Factors specifically implicated at 24 hr post-activation include E2F1, a transcriptional regulator of both cell cycle and apoptosis in T cells (*Gao et al., 2004*; *DeRyckere and DeGregori, 2005*), BHLHE40, a factor known to control multiple aspects of T cell metabolism and differentiation (*Cook et al., 2020*), and the Myc

cofactor MAZ that has not been previously studied in the context of T cell function. Genes connected to factors in the activated T cell networks were enriched for pathways involved in cytokine signaling, the interferon response, transcription, cell cycle, DNA replication and repair, and programmed cell death (*Figure 3—figure supplement 1B*). Together, these data indicate that concurrent but separable stimulation-induced gene programs are the result of the activity of distinct sets of DNA binding factors mobilized by antigen and co-stimulatory receptor signaling in naïve CD4+ T cells.

## Identification of autoimmune variants associated with CD4+ T cell cRE and predicted effector genes

Following our established variant-to-gene (V2G) mapping approach to implicate functional SNPs and their effector genes using the combination of GWAS and chromatin conformation capture data (*Chesi et al., 2019*; *Su et al., 2020*; *Pahl et al., 2021*; *Pahl et al., 2022*; *Su et al., 2022*), we intersected promoter-interacting OCR with autoimmune SNPs from the 95% credible set derived from 15 autoimmune diseases (*Figure 4A*, *Supplementary files 12 and 13*). Constraining the GWAS SNPs in this way reduced the credible set size from an average of 14 variants per sentinel to 3 variants per sentinel. To determine whether open chromatin in physical contact with dynamically regulated genes in CD4+ T cells is enriched for autoimmune disease heritability, we performed a partitioned LD score regression analysis. This landscape was enriched for variants associated with susceptibility to inflammatory bowel disease (IBD), ulcerative colitis (UC), type I diabetes (T1D), lupus (SLE), celiac disease (CEL), allergy (ALG), eczema (ECZ), and rheumatoid arthritis (RA), but not for variants associated with psoriasis (PSO) or juvenile idiopathic arthritis (JIA) (*Figure 4B*, *Supplementary file 14*). The OCR connected to genes upregulated early and/or progressively upon activation (clusters 1 and 3) were most strongly enriched for ALG, CEL, IBD/UC, RA, and T1D heritability, while SLE and ECZ heritability was most enriched in OCR connected to genes upregulated later post-activation (clusters 3 and 5, *Figure 4B*). SLE was also the only disease (besides PSO and JIA) that was not enriched in open chromatin connected to downregulated genes (cluster 4, *Figure 4B*).

The promoter-connected open chromatin landscape for all CD4+ T cell states in this study contains 2606 putatively causal variants linked to approximately half of the sentinel signals (423) for the 15 diseases analyzed, and are in contact with a total of 1836 genes (*Supplementary file 13*). A total of 1151 autoimmune variants localized to the promoters of 400 genes (–1500/+500 bp from the TSS, *Supplementary file 15*). These variants were on average 103 kb from the TSS of their connected gene (*Figure 4C*), and each variant contacted an average of five genes (*Figure 4D*). The majority of linked SNPs interact with genes in addition to the nearest gene, and approximately half of linked SNPs 'skip' the nearest gene to target only distant genes (*Figure 4E*). Approximately 60% of connected genes were implicated across all timepoints (*Figure 4F*, *Supplementary file 13*), while ~40% (753) are dynamically regulated (clusters 1–5) in response to TCR/CD28 co-stimulation. Examples of SNP-genes pairs that exhibit dynamic accessibility, chromosome contact, and expression in response to T cell activation are the *SIK1*, *PARK7*, *DUSP5*, *CLEC2D*, *TRIP10*, *GPR108,* and *IL2* loci (*Figure 4G*, *Supplementary file 13*). The *TRIP10* and *GPR108* promoters were each captured in contact with a high-confidence variant rs1077667 (PP > 0.99), which is located in an intron of *TNFSF14* and is associated with multiple sclerosis (*Figure 4H*). The accessibility of this SNP and its contact with *TRIP10* and *GPR108* increased following activation (*Figure 4H*). Conversely, the allergy-associated SNP rs7380290 is accessible and contacts the *SIK1* promoter in resting cells, but shows reduced accessibility and promoter connectivity upon activation (*Figure 4I*). *TRIP10* encodes a cytoskeletal binding protein involved in endocytosis that suppresses glucose uptake in response to insulin signaling (*Feng et al., 2010*), *GPR108* encodes an orphan G-protein-coupled receptor, and *SIK1* encodes a salt-inducible kinase with roles in cancer, epilepsy, and myeloid signaling (*Darling and Cohen, 2021*). None of these genes have been previously studied in the context of T cell activation.

## Comparative predictive power against orthogonal V2G mapping approaches

In an attempt to gauge the predictive power of our approach relative to other V2G approaches, we compared our chromatin capture-based autoimmune GWAS effector gene predictions to the predictions of four other chromosome capture-based studies in human CD4+ T cells (*Javierre et al., 2016*; *Burren et al., 2017*; *Yang et al., 2020*; *Gate et al., 2018*), four single-cell eQTL studies in human

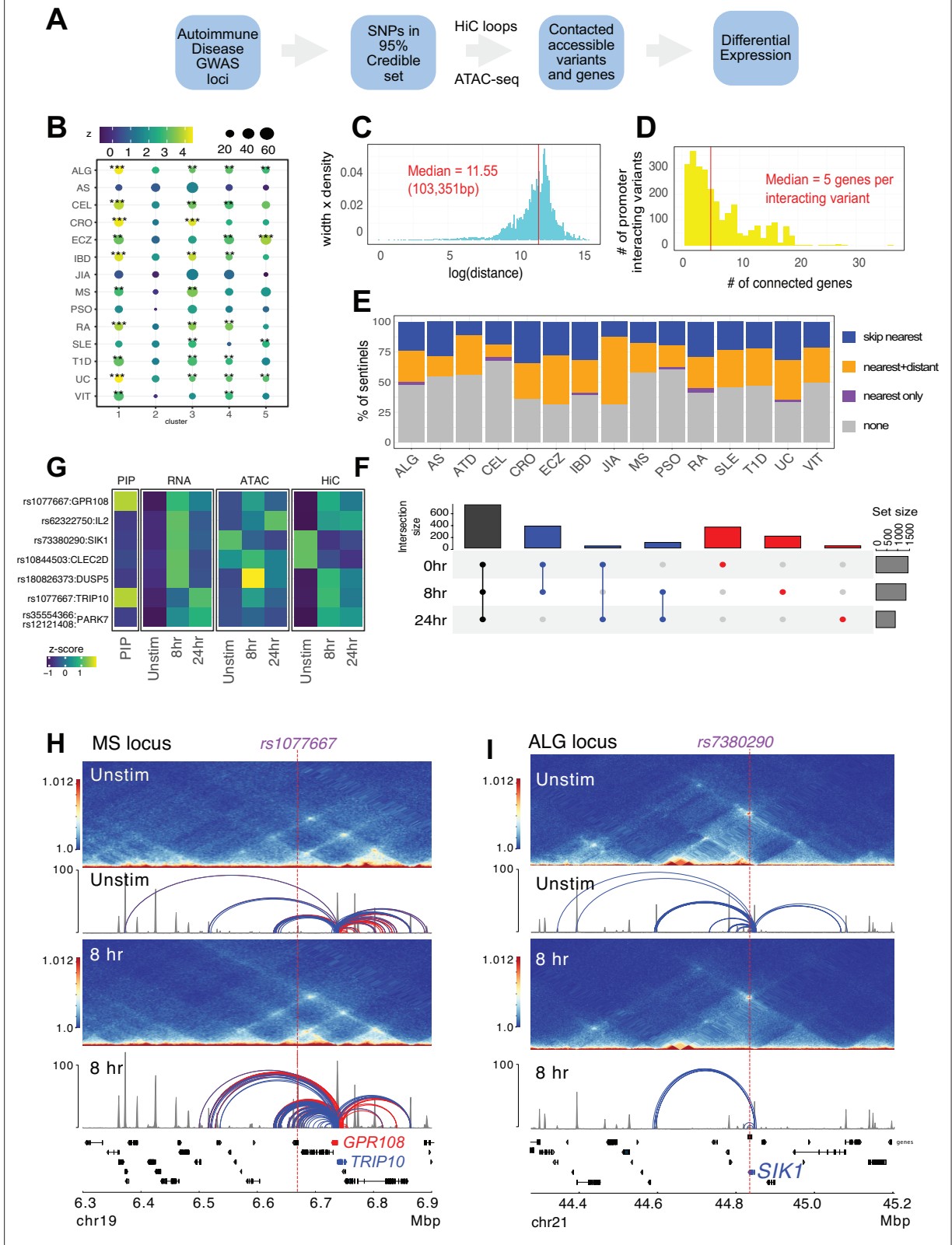

**Figure 4.** Variant-to-gene mapping of autoimmune-associated loci implicates genetic variants in the control of T cell activation. (**A**) Identification of genes in contact with open chromatin regions (OCR) harboring autoimmune-associated SNPs. Sentinel SNPs from 15 autoimmune genome-wide association studies (GWAS) (***Supplementary file 11***) were used to identify the 95% credible set of proxy SNPs for each lead SNP. SNP locations were integrated with ATAC-seq and HiC data to identify the 95% credible set of accessible SNPs in physical contact with a gene promoter. Genes are further

*Figure 4 continued on next page*

*Figure 4 continued*
refined based on expression dynamics over the time course of T cell activation. (**B**) Partitioned LD score regression for autoimmune GWAS using the OCR in contact with the genes in the five clusters defined by RNA-seq. Circle size = enrichment, color = significance, **FDR < 0.01, ***FDR < 0.001. (**C**) Log distribution of the 1D distance between each proxy SNP and its interacting gene based on 3D chromatin conformation (median = 103 kbp). (**D**) Distribution of the number of genes contacted by each accessible variant (median = 5). (**E**) Fraction of open disease-associated variants that interact with no gene promoters (gray), only with the nearest gene promoter (purple), with the nearest gene and a distant gene(s) (orange), and with only a distant gene(s) (blue). (**F**) Number of genes identified by variant-to-gene mapping at each timepoint. Black = shared in all stages, blue = shared in two stages, red = specific to one stage. (**G**) Set of implicated proxy-genes pairs that are both differentially expressed, display differential accessibility, and chromatin contact across T cell activation. (**H**) Example MS variant rs1077667 (p=1.0) exhibits increased accessibility and contact with the promoters of the *GPR108* and *TRIP10* at 8 hr post activation, which are increased in expression at this timepoint. (**I**) Example allergy variant rs7380290 interacts with the *SIK1* gene that is upregulated after activation.

CD4+ T cells (*Soskic et al., 2022*, *Gate et al., 2018*; *Schmiedel et al., 2022*; *Ye et al., 2014*), and a set of 449 genes that when mutated in humans cause inborn errors of immunity (*Tangye et al., 2022*). As part of the comparison, we harmonized the chromatin-based datasets using our chromatin loop calling approach and integrated with the same 95% credible set of autoimmune variants to create comparable physical contact maps. A total of 6936 unique genes were implicated by all studies. Our HiC-ATAC-seq approach in naïve and activated human CD4+ implicated 1947 effector genes, 400 of which through proxies in open promoters and 1836 through contacts with distal variants, of which 752 were dynamically regulated throughout activation. The HiC dataset from Gate et al. implicated 110 genes, and the capture-HiC datasets from Burren et al., Yang et al., and Javierre et al. implicated 1408, 2572, and 2368 genes, respectively (*Supplementary file 16*). A total of 369 autoimmune GWAS effector genes were predicted in common by our HiC approach and the pcHiC-based approaches. The eQTL studies by Soskic et al., Gate et al., Ye et al., and Schmiedel et al. (DICE) implicated 171, 15, 15, and 221 genes, and the co-accessibility data from Gate et al. implicated 45 genes. Concordance between our gene predictions and the pcHiC predictions ranged from 15 to 24% (*Figure 5A*), while concordance between our predictions and the eQTL predictions was low (0.6–5%, *Figure 5A*), and concordance among eQTL studies was also low (1.6–13%, *Figure 5A*). However, when we co-localized the Soskic variant-gene pairs with autoimmune GWAS SNPs, 41 of these eQTL were also predicted by our study, representing a concordance tenfold higher than that obtained by a random sampling of genes within 500 kb of autoimmune variants (*Figure 5B*). A total of 1519 of the genes implicated in our study were not predicted by eQTL approaches, while 644 genes from our CD4 T cell V2G data were not implicated by the other chromatin-based datasets, and altogether 562 putative autoimmune effector genes were uniquely predicted by our study. Potential sources of variation between the results of the studies are outlined in the 'Discussion'. A total of 17 genes were implicated by all CD4+ T cell-based approaches that nominated at least 150 genes (*APIP, BCL2L11, ERAP1, ERAP2, CD5, FIGNL1, IL18R1, METTL21B, MRPL51, PPIF, PTGER4, PXK, RMI2, RNASET2, SERPINB1, TAPBPL, VAMP1*).

To measure the predictive power of the sets of genes implicated by each approach above, a precision-recall analysis was performed against the 'gold standard' human inborn errors in immunity (human inborn errors of immunity [HIEI]) gene set, based on the hypothesis that genes under the control of autoimmune-associated variants would cause monogenic immune disease phenotypes if subjected to deleterious coding mutations. The set of genes from our study with disease-associated SNPs located in their open promoters overlapped with only 5% of the genes from the gold standard set (*Supplementary file 16*, *Figure 5C*). However, leveraging our HiC data to include distal autoimmune variants captured interacting with open gene promoters increased the sensitivity of the approach threefold to 16% (71 overlapping genes, *Figure 5C*), confirming the relevance of long-range promoter–cRE/SNP contacts. This level of sensitivity is comparable to that of other chromosome capture-based datasets for predicting HIEI genes in CD4+ T cells (*Figure 5C*). Conversely, the eQTL-based approaches were 3- to ~12-fold less sensitive than the chromatin-constrained approaches in predicting HIEI genes, with most recalling 0–2.5% and the DICE eQTL dataset recalling 6% (*Figure 5C*). Restricting our 3D chromatin V2G to only those genes dynamically regulated during T cell activation reduced predictive power from 16 to 10%, but the fraction of these genes in the HIEI gold standard set (precision) increased from ~4 to 6.2% (*Figure 5C*), indicating that focusing the V2G set on those SNP-gene pairs that are dynamically regulated by T cell activation reduces potential false-positive predictions. The precision of our dataset was superior to that of the other chromosome capture-based

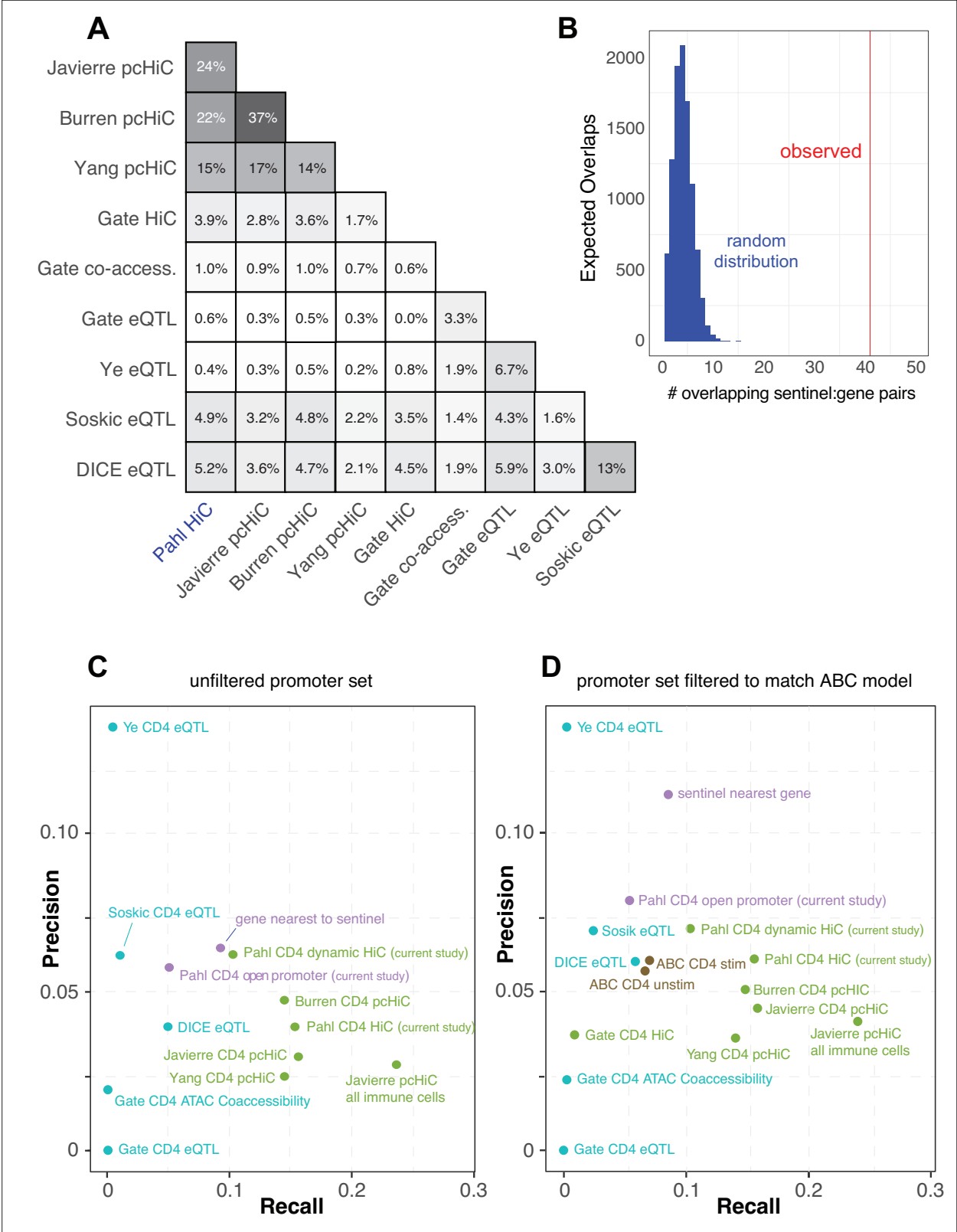

**Figure 5.** Comparative predictive power of orthogonal variant-to-gene (V2G) approaches. (**A**) Percent overlap in predicted genes among chromatin-based and expression quantitative trait mapping (eQTL)-based V2G approaches. (**B**) Physical variant-gene associations (HiC) are enriched for statistical variant-gene associations (eQTL) in activated CD4+ T cells. The histogram depicts the null distribution of shared variant-gene pairs expected at random (~5) while the red line indicates the observed number of variant-gene pairs (41) shared with the 127 eQTL identified by Soskic et al. in a similar CD4+

*Figure 5 continued on next page*

*Figure 5 continued*

T cell activation system. (**C**) Expression of the shared genes. (**D**) Precision-recall analysis of V2G gene predictions against the set of monogenic human inborn errors in immunity.

datasets except for the Burren pcHiC dataset (*Figure 5C*). The eQTL approaches were comparable to the 3D chromatin-based datasets in precision (~3–6%, *Figure 5C*).

We also analyzed the relative predictive power of a recently proposed 'activity-by-contact' (ABC) model (*Nasser et al., 2021*) that uses a multi-tissue averaged HiC dataset instead of cell type-matched 3D chromatin-based data to link variants to genes. This model constrains the input data by removing bidirectional, antisense, and small RNAs, and potentially underweights distal elements connected to ubiquitously expressed genes due to stronger average activity scores near their promoters. Also, while most human genes have multiple alternative transcriptional start sites (median = 3), ABC only annotates only one promoter per gene. To apply the ABC gene nomination model to our CD4+ T cell chromatin-based V2G data, we used our ATAC-seq data and publicly available CD4+ T cell H3K27ac ChIP-seq data as input, and integrated this with GWAS and the averaged HiC dataset from the original ABC study (*Nasser et al., 2021*; *Fulco et al., 2019*). The ABC model nominated 650 genes compared to 1836 genes using our cell type-matched HiC data and analysis pipeline. Only 357 genes were nominated by both approaches, while 1479 genes nominated by our approach were not implicated by the ABC model, and 293 genes not implicated by our approach were newly implicated by ABC. To measure the predictive power of the ABC approach, we re-ran the precision-recall re-analysis with all datasets subjected to the ABC gene-promoter filter (*Figure 5D*). We found that applying the restricted ABC model promoter annotation to all datasets did not have a large effect on recall; however, the precision of several of the datasets was affected. For example, using the restricted promoter/gene set reduced the precision of our V2G approach and artificially inflated the precision of the 'nearest gene to SNP' metric. The precision-recall analysis also shows that the ABC score-based approach is only half as powerful at predicting HIEI genes as the chromatin-based V2G approaches (*Figure 5D*). This indicates that informing GWAS data with cell type- and state-specific 3D chromatin-based data brings more target gene predictive power than application of the multi-tissue-averaged HiC used by the ABC model. Together, these analyses indicate that chromosome capture-based V2G is a more sensitive approach for identifying 'true' effector genes than eQTL or ABC approaches, but comes with additional predicted genes that represent either false positives or true effector genes for which monogenic LOF/GOF mutations have not yet been characterized in humans.

## Functional validation of novel autoimmune V2G-predicted enhancers at the *IL2* locus

Our 3D chromatin-based analysis specifically predicts dynamic, disease-associated regulatory elements in intergenic space at the *IL2* locus. The *IL2* gene encodes a cytokine with crucial, pleotropic roles in the immune system, and dysregulation of IL-2 and IL-2 receptor signaling leads to immunodeficiency and autoimmune disorders in mice and humans (*Cerosaletti et al., 2013*; *Spolski et al., 2018*; *Abbas et al., 2018*; *Joosse et al., 2021*). Activation-induced transcription of *IL2* involves an upstream regulatory region (URR) ~375 bp from the TSS that has served as a paradigm of tissue-specific, inducible gene transcription for nearly four decades (*Fujita et al., 1986*; *Durand et al., 1987*; *Novak et al., 1990*). However, the presence of evolutionarily conserved non-coding sequences (CNS) in the ~150 kb of intergenic space 46, 51, 80, 83, 85, 96, 122, and 128 kb upstream of the TSS suggests that additional regulatory elements may have evolved to control *IL2* (*Mehra and Wells, 2015*; *Figure 6A*). The –51 kb CNS contains an SNP linked to T1D, IBD, PSO, CEL, and allergy (rs72669153), while the –85 kb CNS contains an SNP linked to RA (rs6232750) and the –128 kb CNS contains two SNPs linked to T1D, JIA, and SLE (rs1512973 and rs12504008, *Figure 6A*). In TCR/CD28-stimulated naïve CD4+ T cells, these CNS are remodeled to become highly accessible (*Figure 6A*), and they loop to interact physically with the *IL2* URR (*Figure 6B*) at both timepoints. ChIP-seq analyses in human T cells (*Consortium, 2004*) show that the URR and all distal CNS except –85 are occupied by TF such as Jun/Fos (AP-1), NFAT, and NFkB that are known regulators of *IL2* (*Figure 6—figure supplement 1*), and the –85,–122, and –128 CNS are occupied by additional TF not previously thought to be direct regulators of *IL2*, such as MYC, BCL6, and STAT5 (*Figure 6—figure supplement 1*). Recombinant reporter assays in primary activated human CD4+ T cells showed that the –46,–51, –83, and –128 CNS/OCR can enhance transcription

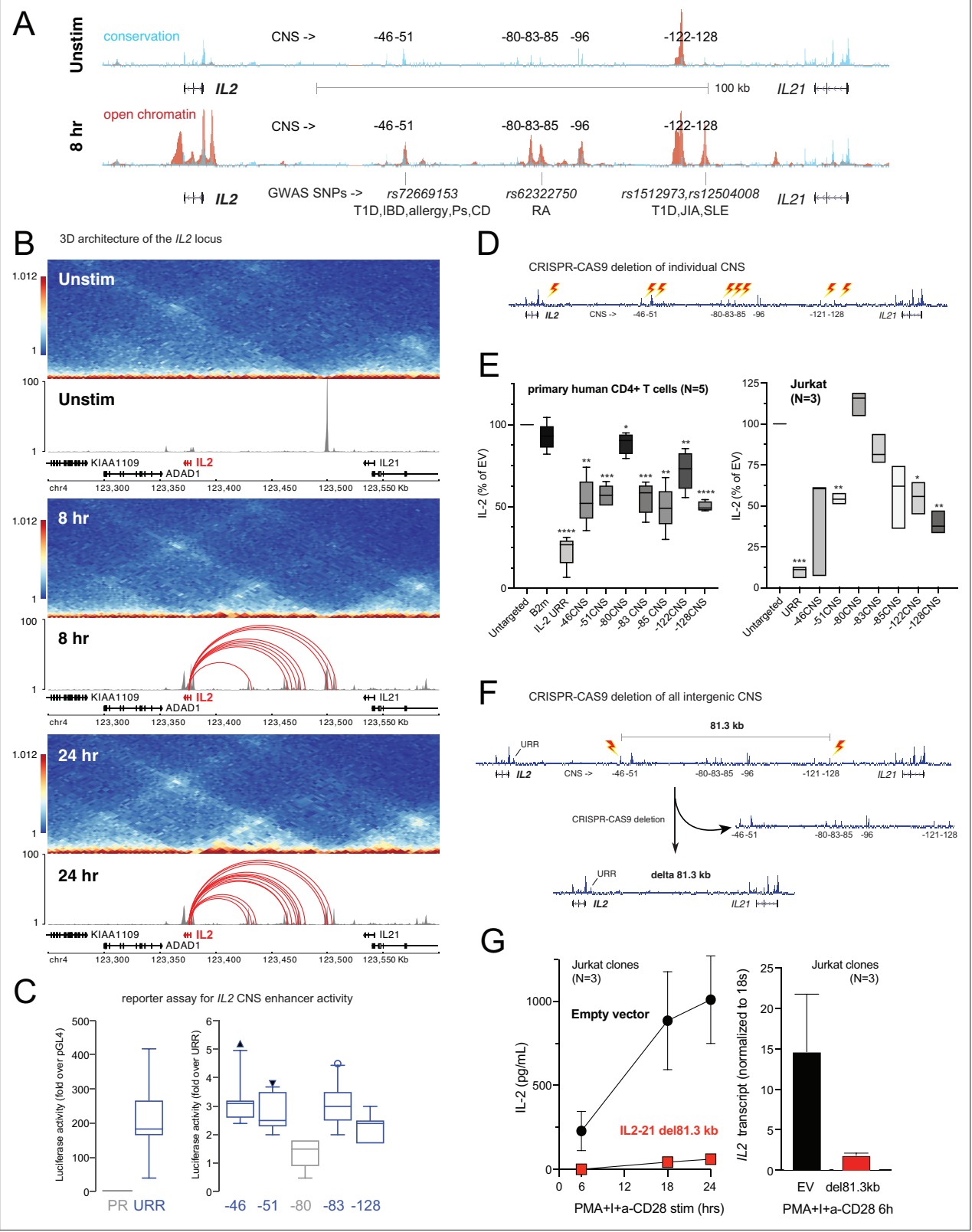

**Figure 6.** Functional validation of autoimmune variant-to-gene (V2G)-implicated *cis*-regulatory elements (cRE) at the *IL2* locus. (**A**) The combination of evolutionary conservation (blue), open chromatin (red), and autoimmune disease-associated SNPs at the *IL2* locus identify putative cRE in quiescent vs. 8 hr activated naïve CD4+ T cells. (**B**) Activation-dependent topologically associating domains (TAD)/sub-TAD structure (heatmaps), chromatin remodeling (gray), and promoter–open chromatin region (OCR) interactions (red) at the *IL2* locus. (**C**) Recombinant reporter assay showing

*Figure 6 continued on next page*

*Figure 6 continued*

transcriptional activity of the *IL2* upstream regulatory region (URR) (+35 to −500 from the TSS) in activated primary CD4+ T cells (N = 7 donors) compared to a basal promoter (pGL4, left panel). The right panel depicts transcriptional activity of the CNS regions indicated in (**A**) cloned upstream of the URR. All regions except the −80 CNS show statistically significant activity relative to the URR alone (N = 7, p<0.05, line = median, box = 95/5% range). (**D**) Scheme of CRISPR/CAS9-based deletion of individual *IL2* CNS using flanking gRNAs. (**E**) Activation-induced secretion of IL-2 protein by CRISPR-targeted primary CD4+ T cells (N = 5 donors, left panel) or Jurkat cells (N = 3 replicates, right panel) relative to untargeted control (CAS9, no gRNA) cells (****p<0.0001, ***p<0.001, **p<0.01, *p<0.05, line = median, box = 95/5% range). In primary cells, *B2M* gRNAs served as an irrelevant targeted control. (**F**) Scheme of CRISPR/CAS9-based deletion of the 81.3 kb region containing all distal *IL2* cRE using flanking gRNAs in Jurkat cells. (**G**) Activation-induced IL-2 protein (left panel) and mRNA (right panel) by control (black) vs. 81.3 kb deleted (red) Jurkat cells (N = 3 separate clones).

The online version of this article includes the following figure supplement(s) for figure 6:

**Figure supplement 1.** Transcription factor (TF) occupancy and stability of distal *IL2 cis*-regulatory elements (cRE) in CD4+ T cell subsets.

from the URR (*Figure 6C*). To determine whether the native elements contribute to the expression of *IL2*, we targeted each CNS/OCR individually in primary human CD4+ T cells or Jurkat T cells using CRISPR/CAS9 (*Figure 6D*) and measured IL-2 secretion following TCR/CD28 stimulation (*Figure 6E*). Deletion of the −46,−51, −83, −85, −122, and −128 kb elements in primary human CD4+ T cells each resulted in an ~50% reduction in IL-2 production, while deletion of the −80 kb element had little effect. A very similar pattern of impact was observed when these elements were deleted individually in Jurkat T cells (*Figure 6E*). The URR has a stronger contribution to IL-2 production than any individual intergenic element, as deletion of the URR almost completely abrogated activation-induced IL-2 production by both primary CD4+ or Jurkat T cells (*Figure 6E*). To determine whether these intergenic enhancers exist in synergistic epistasis (*Lin et al., 2022*) necessary for *IL2* transactivation, we generated Jurkat T cell clones in which the stretch of all seven elements located 46–128 kb upstream of *IL2* was deleted using CRISPR/CAS9 (*Figure 6F*). Despite the URR and 46 kb of upstream sequence being intact in these clones, loss of the 81.3 kb stretch of intergenic enhancers renders these cells incapable of expressing *IL2* at both the mRNA and protein levels in response to stimulation (*Figure 6G*). These results show that the URR is not sufficient for activation-induced expression of *IL2*, and that *IL2* has a previously unappreciated, complex, and autoimmune disease-associated regulatory architecture that was accurately predicted by our 3D epigenomic V2G approach. Importantly, we find that the distal *IL2* cRE are highly accessible in quiescent memory T cell subsets (Th1, Th2, Th17, Th1-17, Th22) isolated directly ex vivo from human blood, whereas naïve CD4+ T cells and non-IL-2-producing Treg showed little accessibility at these elements (*Figure 6—figure supplement 1D*). This suggests that stable remodeling of distal *IL2* cRE can persist in vivo after TCR signals cease, and that this epigenetic imprinting contributes to the immediate, activation-induced production of IL-2 exhibited by memory, but not naïve or regulatory, CD4+ T cells (*Hedlund et al., 1989*; *Picker et al., 1995*; *Sojka et al., 2004*).

## Distal *IL2* enhancers are evolutionarily conserved and impact in vivo T cell-mediated immunity in mice

The intergenic *IL2* enhancers defined above are conserved at the sequence level between human and mouse (*Figure 7A*). To test whether enhancer function is likewise evolutionarily conserved, we used zygotic CRISPR/CAS9 targeting to generate mice with a 583 bp deletion of the ortholog of the −128 kb human enhancer CNS/OCR situated 83 kb upstream from mouse *Il2* (*Il2*-83-cRE-ko, C57BL6 background). Deletion of this genomic region did not discernibly affect T lymphocyte development in *Il2*-83-cRE-ko mice. However, peripheral CD4+ T cells from *Il2*-83-cRE-ko mice produced substantially less IL-2 at both the protein and mRNA levels, and exhibited reduced proliferation, in response to in vitro stimulation (*Figure 7B*), confirming that the enhancer function of the orthologous −128 and −83 kb elements is conserved between human and mouse. The in vitro induction of Foxp3+ regulatory T cells (Treg) from conventional CD4+ T cell precursors in response to antigenic stimulation in the present of TGF-β, a differentiation process highly dependent upon IL-2, was reduced fourfold in *Il2*-83-cRE-ko mice compared to wild-type mice (*Figure 7C*), but could be rescued by the addition of exogenous IL-2 to the culture (*Figure 7C*).

To test whether the distal −83 kb *Il2* enhancer contributes to physiological immune responses in vivo, we immunized wild-type and *Il2*-83-cRE-ko mice with the model antigen chicken ovalbumin. *Il2*-83-cRE-ko animals showed a nominal increase in the differentiation of CD4+ T cells into CXCR5+PD-1[hi]

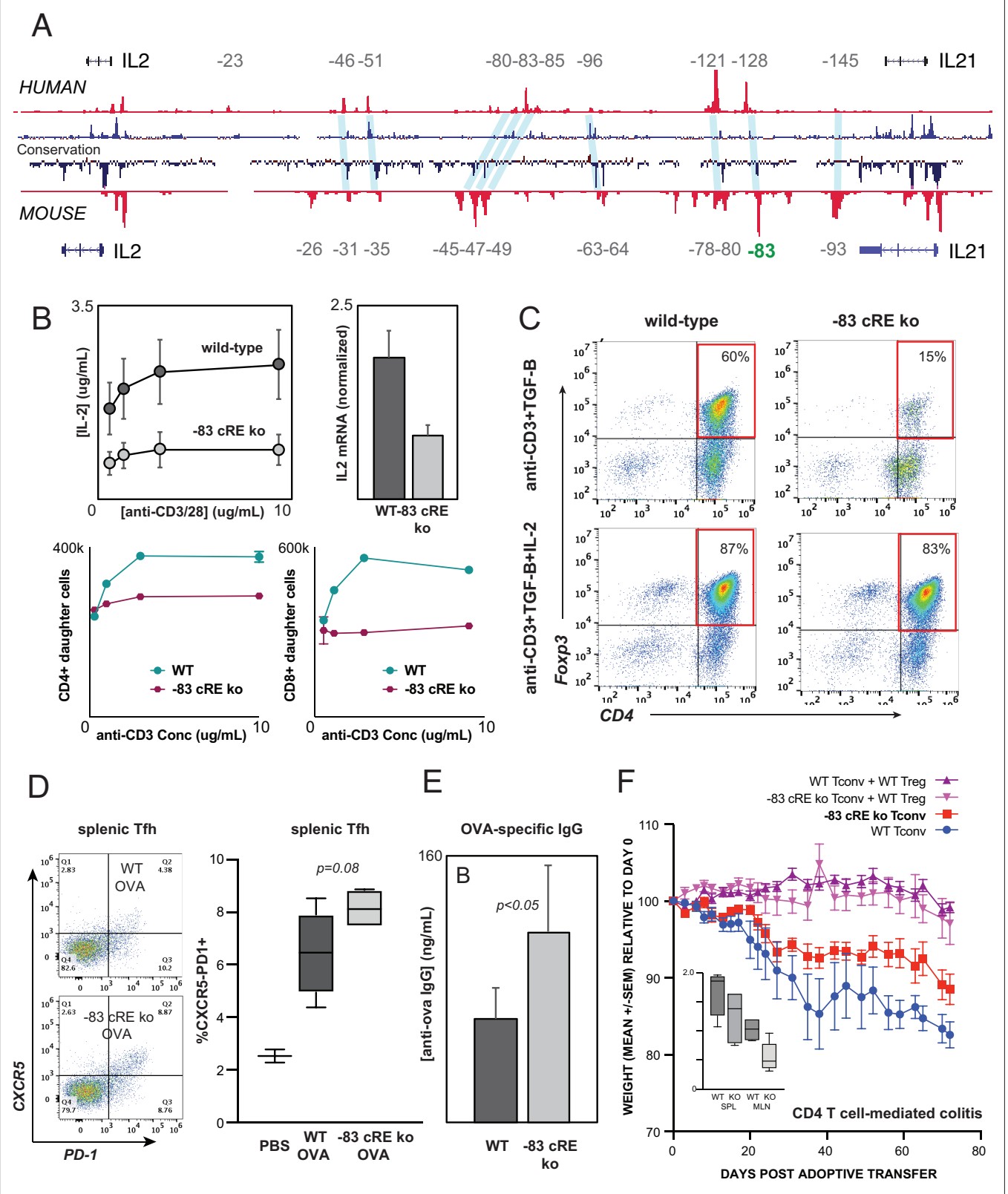

**Figure 7.** Distal *IL2* enhancers are evolutionarily conserved and impact in vivo T cell-mediated immunity in mice. (**A**) Map of the human and mouse *IL2* locus depicting mammalian conservation (dark blue), ATAC-seq from activated human and mouse CD4+ T cells (red), and orthologous non-coding sequences conserved between mouse and human (light blue). (**B**) IL-2 protein secretion (top-left panel) and *Il2* mRNA expression (top-right panel) by CD4+ T cells from wild-type or *Il2*-83 *cis*-regulatory elements (cRE) ko mice in response to stimulation with plate-bound anti-CD3 and anti-CD28

*Figure 7 continued on next page*

*Figure 7 continued*

Ab in vitro. Bottom panels show soluble anti-CD3-induced in vitro clonal expansion of wild-type or *Il2*-83 cRE ko CD4+ (left) and CD8+ (right) T cells measured by dye dilution (*Wells et al., 1997*). (**C**) Foxp3 expression by murine CD4+CD25- Tconv stimulated with anti-CD3 Ab and TGF-β in the absence (top panels) vs. presence (bottom panels) of exogenous IL-2. Wild-type or *Il2*-83 cRE ko mice (N = 6) were immunized intraperitoneally with chicken ovalbumin in incomplete Freund's adjuvant. The frequency of CD4+PD1+CXCR5hi follicular helper T cells in the spleens of three animals was measured by flow cytometry at day 5 post-immunization (**D**), and levels of ovalbumin-specific IgG in the serum of three animals were measured at day 10 post-immunization (**E**). (**F**) In vivo inflammatory bowel disease (IBD) model. CD4+CD25- Tconv from wild-type (pink, red) or *Il2*-83 cRE ko (purple, blue) mice were transferred alone (purple, pink) or together with wild-type CD4+CD25+Treg (red, blue) into RAG1-ko mice (N = 5 per group). Animal weight was monitored for 40 days, and the number (in millions) of activated CD4+Tconv in the spleens and mesenteric lymph nodes was measured by flow cytometry (inset). Statistical significance (*p<0.05) was measured by ANOVA or *t*-test.

follicular helper T cells (Tfh) in the spleen compared to wild-type animals (*Figure 7D*), a process known to be antagonized by IL-2, and generated significantly elevated levels of ovalbumin-specific IgG antibody following immunization (*Figure 7E*). To determine whether the 83 kb *Il2* enhancer contributes to auto-inflammatory disease susceptibility in vivo, we used an adoptive transfer model of IBD in which the development of colitis is determined by the balance between conventional and regulatory CD4+ T cell activities (*Powrie et al., 1994*; *Morawski et al., 2013*). Upon transfer into lymphocyte-deficient Rag1-ko recipients, wild-type CD4+T cells respond to microbiota in the gut by inducing inflammatory colitis that leads to weight loss (*Figure 7F*, WT Tconv), while co-transfer of wild-type Treg effectively controls inflammation (*Figure 7F*, WT Tconv+WT Treg). Transfer of *Il2*-83-cRE-ko Tconv led to significantly less colitis (*Figure 7F*, *Il2*-83-cRE-ko Tconv) and was associated with a nominal decrease in *Il2*-83-cRE-ko Tconv in the spleen and a significant decrease in the accumulation of *Il2*-83-cRE-ko Tconv in the mesenteric lymph nodes (MLN) that drain the intestines (*Figure 7F*, inset). Treg require IL-2 from Tconv for their function and homeostasis (*Smigiel et al., 2014*), and despite their reduced numbers, *Il2*-83-cRE-ko Tconv were able to support the homeostasis and function of co-transferred wild-type Treg (*Figure 7F*, *Il2*-83-cRE-ko Tconv+WT Treg). The −83 kb ortholog of the human −128 kb *IL2* enhancer contributes to physiological immune responses and inflammatory disease susceptibility in vivo.

## Impact of autoimmune risk-associated genetic variation on CRE activity

The chromatin conformation approach used here employs GWAS variants as 'signposts' to identify disease-relevant regulatory elements and connect them to their target genes, but does not per se determine the effect of disease-associated genetic variation on enhancer activity or gene expression. We experimentally measured variant effects at the *IL2* locus, where the −128 enhancer defined above contains two SNPs linked to T1D, JIA, and SLE (rs1512973 and rs12504008). Using a recombinant reporter assay in primary activated CD4+ T cells, we confirmed that disease-associated genetic variation influences intergenic *IL2* enhancer activity at the −128 kb element, in that the risk allele contributes significantly less transcriptional activity than the reference allele (*Figure 8A*).

Autoimmune variants are likely to influence disease risk by altering the activity of CRE in T cells. At genome scale, we identified over 1000 cRE likely impacted by autoimmune disease-associated genetic variation, in that they contain autoimmune risk SNPs that are predicted to decrease or increase TF binding affinity. Overall, 1370 autoimmune risk variants in open chromatin were predicted to influence the activity of 495 DNA binding factors (*Supplementary file 17*), including PLAG1, PRDM1, BACH2, MYC, TBX21, BHLHE40, LEF1, TCF7, BCL6, IRF, p53, STAT (*Figure 8—figure supplement 1A*), and hundreds of NFkB, EGR, KLF, FKH/FOX, and FOS-JUN sites (*Figure 8B*). For example, the T1D SNP rs3024505 (p=0.200) connected to the promoters of *IL9* and *FAIM3* (*Figure 8—figure supplement 1B*) and the celiac SNP rs13010713 (p=0.154) contacting the *ITGA4* promoter (*Figure 8—figure supplement 1C*) are predicted to disrupt binding sites for *MZF1* (*Figure 8—figure supplement 1D*) and SOX4 (*Figure 8—figure supplement 1E*). Similarly, the MS SNP rs1077667 contacting the promoters of *GPR108* and *TRIP10* (*Figure 4H*) is predicted to reduce affinity for TP53, TP63, and OCT2/POU2F2 (*Figure 8—figure supplement 1F*).

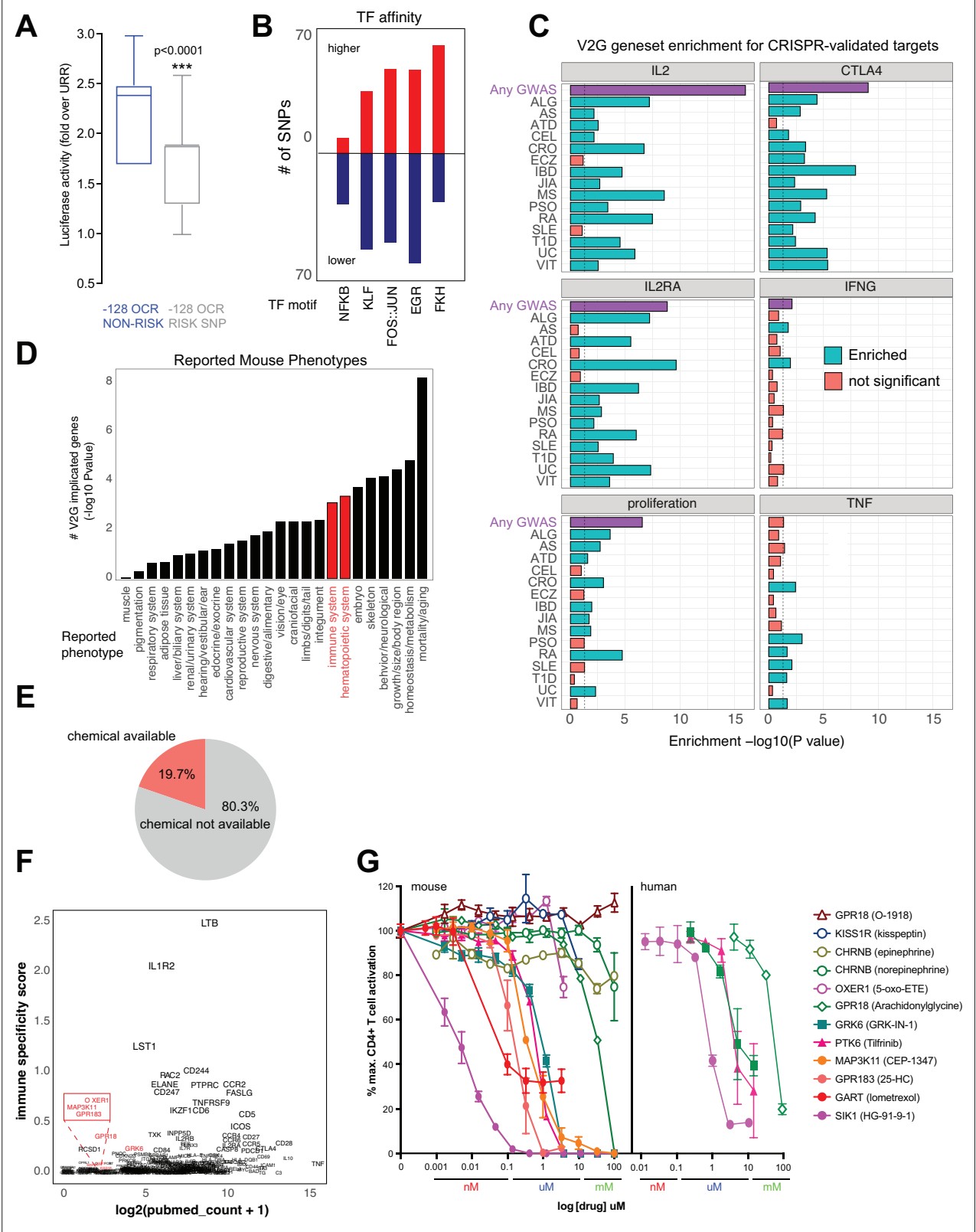

**Figure 8.** Functional validation of autoimmune variant-to-gene (V2G)-implicated genes. (**A**) Recombinant reporter assay in primary activated CD4+ T cells (N = 7 donors) showing transcriptional activity of the reference vs. risk alleles of the *IL2* –128 *cis*-regulatory elements (cRE) relative to the upstream regulatory region (URR) alone (p<0.0001). (**B**) Prominent transcription factor (TF) motifs predicted to be disrupted (blue) or stabilized (red) by promoter-connected autoimmune SNPs. (**C**) 3D chromatin-based V2G genes are enriched for CRISPR-implicated genes that regulate CD4+ T cell activation.

*Figure 8 continued on next page*

*Figure 8 continued*

Observed enrichment of genes regulating multiple aspects of CD4+ T cell activation (IL-2, IL-2 receptor, CTLA-4, IFNg, TNFa or proliferation) from CRISPR screens by Freimer, Schmidt, and Shifrut among sets of 3D chromatin-implicated genes among individual diseases (green, FDR < 0.05) or all diseases (purple, FDR < 0.05). (**D**) Enrichment for 3D chromatin-based autoimmune V2G genes among genes with germline knock-out mouse immune (red) and other (black) phenotypes (adjP < 0.05, International Mouse Phenotype Consortium [IMPC] database). (**E**) Autoimmune V2G-implicated genes with at least one pharmacological modulator (rDGIdb). (**F**) Comparison of the number of manuscripts retrieved from PubMed related to autoimmune disease for each V2G gene with pharmaceutical agents available (x-axis) with an immune-specific expression score computed using the sum GTEX median expression values (v8) for whole blood or spleen divided by other tissues (y-axis). Genes highlighted in red were selected for functional validation in (**G**). (**G**) Dose-dependent impact of the indicated pharmacological agents targeting the V2G-implicated genes *KISS1R, CHRNB, OXER1, GPR18, GRK6, PTK6, MAP3K11, GPR183, GART,* and *SIK1* on proliferation of anti-CD3/28 activated murine (left panel) or human (right panel) CD4+ T cells in vitro (N = 4).

The online version of this article includes the following figure supplement(s) for figure 8:

**Figure supplement 1.** Predicted impact of autoimmune disease-associated genetic variation at variant-to-gene (V2G)-implicated loci.

**Figure supplement 2.** Orthogonal validation of 3D chromatin-based variant-to-gene (V2G) genes.

## Functional validation of chromosome capture-based V2G-implicated effector genes

To determine whether genes identified via their physical interaction with autoimmune variants in CD4+ T cell chromatin contact maps tend to be directly involved in T cell activation and function, we compared the set of autoimmune genes implicated by chromatin contacts in this study to sets of genes identified in CRISPR-based screens that control aspects of CD4+ T cell activation like proliferation and expression of the inflammatory genes *IFNG, CTLA4, IL2, IL2RA,* and *TNF* (*Shifrut et al., 2018*; *Schmidt et al., 2022*; *Freimer et al., 2022*). The set of all V2G-implicated genes was highly enriched for genes shown to regulate IL-2, IL-2 receptor, CTLA-4, and proliferation (*Figure 8C, Figure 8— figure supplement 2A, Supplementary file 18*). For example, 202 genes shown to regulate IL-2 production and 166 genes shown to regulate proliferation were also implicated in our autoimmune V2G set (*Figure 8—figure supplement 2A, Supplementary file 16*). Genes implicated by V2G in activated CD4+ T cells were moderately enriched for genes known to control the production of IFN-G, but at the individual disease level, only genes connected to CRO- and ATD-associated variants were enriched for IFNG regulatory genes (*Figure 8C, Figure 8—figure supplement 2A*). Genes contacting CRO, PSO, RA, SLE, T1D, and VIT variants were moderately enriched for TNF regulatory genes, but the set of all V2G genes was not enriched for TNF genes. We also queried the orthologs of our V2G-implicated genes from the International Mouse Phenotype Consortium (IMPC) database and identified 97 genes that when knocked out give an immune phenotype and 126 V2G genes that result in a hematopoietic phenotype (*Figure 8D, Figure 8—figure supplement 2B, Supplementary file 18*). This frequency of observed immune/hematopoietic phenotypes represents a significant (adjP < 0.05) ~30% enrichment over expected. This gene set was also enriched for mortality, homeostasis/metabolism, growth/body size, skeleton, and embryonic phenotypes (*Figure 8D, Supplementary file 18*).

An important application of this V2G approach is the identification of novel regulators of T cell activation and their potential as drug targets as nearly 20% of implicated genes have at least one chemical modulator currently available (*Figure 8E, Supplementary file 19*). As shown above, this approach validates genes that are well-studied regulators of T cell function; however, a significant portion of implicated genes are not well-studied and are not currently known to regulate T cell activation (*Figure 8F*). We observed a trend that genes expressed more highly in immune tissues (GTEX) have in general been better investigated and identified several less-studied genes that could be novel targets, including *GRK6, PTK6, SIK1,* and *MAP3K11* encoding kinases, *OXER1, GPR183, GPR18,* and *KISS1R* encoding G-protein-coupled receptors, CHRNB1 encoding an acetylcholine receptor, and *GART* encoding a de novo purine pathway enzyme. To determine whether these V2G-implicated genes are novel regulators of T cell activation, we used commercially available pharmacological modulators in dose–response assays of activation-induced T cell proliferation. Stimulation of T cells in the presence of ligands for CHRNB1, KISS1R and OXER1 did not significantly affect T cell proliferation; however, small molecules targeting GRK6, PTK6, MAP3K11, GPR183, GART, and SIK1 inhibited T cell activation in the nanomolar to micromolar range (*Figure 8G*). Together, these data show that maps of dynamic, 3D variant-gene chromatin contacts in stimulated CD4+ T cells are able to identify genes with bona fide roles in T cell activation.

## Discussion

By measuring dynamic changes in chromosome folding, chromatin accessibility, and gene expression in naïve CD4+ T cells as a function of TCR-CD28 co-stimulation, we identified the putative *cis*-regulatory landscape of autoimmune disease-associated genetic variation and physically connected these elements to their putative effector genes. This and prior chromosome capture-based studies show that most cRE and their variants interact with, and therefore have the potential to regulate, more than one gene (median 5 in this study), supporting a scenario in which multiple effector genes are operative at a GWAS locus. We validated a stretch of novel distal elements predicted by our V2G approach as bona fide enhancers for the canonical immune gene *IL2* and showed that autoimmune-associated genetic variation at one of these elements influences its activity. We also conducted pharmacological targeting experiments and compared our results with CRISPR-based studies to validate sets of bona fide effector genes with a confluence of multiple orthogonal lines of evidence supporting their role in CD4+ T cell activation.

We also observe that the vast majority (87–95%) of GWAS effector genes predicted by chromosome capture-based approaches are not the genes nearest to the sentinel SNPs queried in this study, while roughly one-third of eGenes predicted by eQTL approaches are the nearest to a sentinel. This and the low concordance between 3D chromatin vs. eQTL eGene predictions is consistent with the view that eQTL-based and GWAS-based approaches inherently implicate different types of genes. While eQTLs cluster strongly near transcription start sites of genes with simple regulatory structures and low enrichment for functional annotations, GWAS variants are generally far from the TSS of genes with complex regulatory landscapes (*Mostafavi et al., 2023*). Moreover, eGene discovery by eQTL studies using scRNA-seq approaches are significantly limited by the low gene detection power inherent to these methods, particularly in rare cell types.

Our systematic comparison of 3D chromatin-based target gene nomination studies in human CD4+ T cells revealed significant variability between datasets, with the highest concordance exhibited by the Javierre and Burren datasets (37%) and the next highest exhibited by our V2G and the Javierre dataset (24%). Although we harmonized the comparisons as much as possible by subjecting each dataset to the same HiC loop calling and GWAS integration steps, there are several potential sources for the observed discrepancy between the studies. The modes of stimulation are largely comparable, but timepoints and donors varied, and ours was the only study that sorted naïve CD4+ T cells prior to stimulation. The higher concordance among promoter-capture datasets compared to our HiC dataset is likely due to their lower resolution compared to our HiC and their greater sequencing depth focused at promoters compared to HiC. The lower resolution of HindIII-based capture-HiC results in loops called to the wrong promoters (*Su et al., 2021b*) and will miss distal SNP interactions at any promoters excluded from the capture set. While HiC is unbiased in this regard, high-resolution HiC will fail to call some SNP-promoter loops because the sequencing depth is spread across the whole genome instead of focused at promoters. However, despite variation between studies, the results show the clear value of tissue-matched chromatin conformation maps compared to tissue-averaged HiC for understanding the complex genetic and epigenetic mechanisms that regulate gene expression and for predicting autoimmune effector genes.

Why do the V2G approaches not capture a larger proportion of genes from the human inborn errors in immunity 'truth set'? Low recall/sensitivity could result from the fact that a substantial portion of the mutations in the full HIEI gene set result in immunodeficiency but not autoimmune phenotypes. However, restricting the HIEI truth set to only those disease gene mutations that result in an autoimmune phenotype (143 genes) only increased recall from 16 to 17.5%, suggesting that this is not a major factor reducing sensitivity. Also, our method uses GWAS signals as an input, and unlike GWAS for height, body mass index, blood traits, etc., most autoimmune GWAS signals are likely not saturated, which limits discoverability. Another reason for reduced sensitivity is the likelihood that many of the genes in the HIEI truth set operate in cells other than CD4+ T cells, and consistent with this, when the Javierre pcHiC V2G is extended to all immune cell types tested, recall increases from 17 to 27%. This observation emphasizes how limited inclusion cell types and states in a study can significantly limit the power to detect autoimmune effector genes. Why are so few of our 3D chromatin-based V2G genes present in the HIEI truth set? Low precision could be due to false positives in the V2G gene set; that is, contacts between disease-associated cRE and genes that are not in fact involved in disease susceptibility. This does not argue per se against the biological relevance of the cRE or

the gene, only that the linkage to disease susceptibility is not relevant. Alternatively, the absence of chromatin-GWAS-implicated genes in the truth set could be a false negative from the point of view of the truth set; that is, monogenic, disease-causing mutations in these genes may exist in the human population, but have not yet been discovered. For example, in the year 2000 approximately 100 monogenic mutations causing human inborn errors of immunity were known, but this increased by ~11 disease mutations per year until 300 human inborn errors of immunity had been identified by 2017. The advent of next-generation exome/genome sequencing and the COVID-19 pandemic resulted in an increase in the rate of HIEI discovery between 2018 and 2021 to ~40 per year to the current recognized HIEI of ~450. If we assume a conservative continued rate of HIEI discovery of 25 disease genes per year, the next 10 years will show us an additional ~250 disease genes that are not currently contained in the truth set.

Many of the genes identified in this 3D epigenomic V2G screen have known roles in T cell activation and function. An example is *IL2*, and we used the resulting maps to identify and validate a stretch of previously unknown distal enhancers whose activity is required for *IL2* expression and is influenced by autoimmune genetic variation. Another example is the phosphatase DUSP5 that regulates MAPK signaling during T cell activation (*Kovanen et al., 2008*; *Chuang and Tan, 2019*). However, roles for many of the genes implicated here in T cell activation are not known. For example, one of the top implicated genes, PARK7, is a deglycase studied in the context of Parkinson's disease, but has recently been shown to modulate regulatory T cell function (*Danileviciute et al., 2022*). The orphan G-protein-coupled receptor GPR108 is another top gene uniquely implicated by our chromatin-based V2G that has not been studied in T cells, but was identified in a recent CRISPR screen for genes affecting IL-2 levels (*Freimer et al., 2022*). Also co-implicated by our study and recent CRISPR screens are the cannabidiol receptor GPR18 (*McHugh et al., 2010*) and the purine biosynthetic enzyme GART (*Kan et al., 1993*). The GPR18 agonist arachidonyl glycine inhibited CD4+ T cell activation above 10 uM, while the GPR18 antagonist O-1918 slightly enhanced T cell activation. This GPR was implicated by both chromatin- and eQTL-based approaches. Antagonism of GART, an enzyme we previous identified as a V2G effector gene in COVID19 severity (*Pahl et al., 2022*), with the FDA-approved drug lometrexol inhibited T cell activation in the 10 nM range. Antagonism of GRK6, a member of the G-coupled receptor kinase family associated with insulin secretion and T2D susceptibility (*Varney et al., 2022*), and PTK6, an oncogenic kinase studied in the context of cancer (*Gilic and Tyner, 2020*), led to the inhibition of T cell activation in the nM to uM range. These targets were implicated by the other chromatin-based approaches but not by eQTL. Inhibition of MAP3K11, a kinase that facilitates signaling through JNK1 and IkappaB kinase in cancer (*Slattery et al., 2012*; *Knackmuss et al., 2016*), inhibited stimulation-induced CD4+ T cell proliferation in the 100 nM range, as did 25-hydroxycholesterol, a ligand of the G-protein-coupled receptor GPR183 that was implicated by both chromatin- and eQTL-based approaches. SIK1 is a member of the salt-inducible kinase family uniquely implicated in our study that negatively regulates TORC activity (*Darling and Cohen, 2021*), and a small-molecule SIK1 inhibitor potently antagonized stimulation-induced CD4+ T cell activation in the pM range.

Our integration of high-resolution Hi-C, ATAC-seq, RNA-seq, and GWAS data in a single immune cell type across multiple activation states identified hundreds of autoimmune variant-gene pairs at approximately half of all GWAS loci studied, and application of this technique to additional immune cell types will likely identify effector genes at many of the remaining loci. This study highlights the value of chromosome conformation data as a powerful biological constraint for focusing V2G mapping efforts (*Hammond et al., 2021*) and shows that dynamic changes in the spatial conformation of the genome that accompany cell state transitions alter gene expression by exposing promoters to a varying array of CRE, TF, and genetic variants.

# Materials and methods
## T cell isolation and in vitro stimulation
Human primary CD4+ T cells were purified from the apheresis products obtained from healthy, screened human donors through the University of Pennsylvania Human Immunology Core (HIC) under approved IRB protocol 20-017313. Naïve CD4+ T cells were purified using EasySep™ human naïve CD4+ T cell isolation kit II (STEM Cells Technologies, Cat# 17555) by immunomagnetic negative selection as per the manufacturer's protocol. Isolated untouched, highly purified (93–98%) naïve human

CD4 T cells were activated using anti-CD3+anti-CD28 Dynabeads (1:1, Thermo Fisher Scientific, Cat# 11161D) for 8–24 hr. Cells were then used to prepare sequencing libraries. The human leukemic T cell line Jurkat was obtained from ATCC, cloned by limiting dilution, and clones with high activation-induced secretion of IL-2 were selected for further study. Single-cell mouse lymphocyte suspensions were prepared from spleen and lymph nodes isolated from 6-week-old female wild-type or *Il2*-83cRE-ko mice (C57BL6 background) housed in the fully AAALAC-accredited CHOP vivarium under a CHOP IACUC-approved protocol 22-000594. Mouse CD4+ T cells were purified by negative selection using a CD4+ T cell isolation kit (Miltenyi Biotec, Cat# 130-104-454). For CD8 depletion, CD8+ T cells were positively stained using CD8a (Ly-2) microbeads (Miltenyi Biotec, Cat# 130-117-044) and the negative flow-through fraction was collected. Tregs were purified from the mouse lymphocytes using a CD4+CD25+regulatory T cell isolation kit (Miltenyi Biotec, Cat#130-091-041). The purity of the isolated cells was checked by flow cytometry, showing approximately 95% cell purity. For iTreg induction, CD4+CD25 T cells (1 × 10⁶) were activated in a 24-well plate pre-coated with anti-CD3 (1 µg/mL) and the cells were cultured for 72 hr in RPMI culture medium with soluble anti-CD3 (0.5 µg/mL), IL-2 (25 units/mL), TGF-β (3 ng/mL), anti-IFN-γ (5 µg/mL), and anti-IL-4 (5 µg/mL). iTreg induction cultures were also set up without adding IL-2. iTreg cultures were harvested after 72 hr, and cells were stained for Foxp3 expression.

## *Il2*-83-cRE ko mice

The CRISPR/CAS method was employed to delete the intergenic −83 CNS sequence between *Il2* and *Il21*. CRISPR guide RNAs were designed (two guide RNAs upstream of −83CNS) and one guide RNA (downstream of −83CNS) using guide RNA design tools (http://crispr.mit.edu.guides). The pX335 plasmid (Addgene #42335) was used to generate a DNA template for sgRNA in vitro transcription. To in vitro transcribe the sgRNA, a T7 sequence was incorporated at the 5' end of the guide RNA sequence, and a part of the sgRNA scaffold sequence from px335 was added at the 3' end of the guide RNA. Oligonucleotide sequences containing the guide RNA (underlined), along with these added sequences, were synthesized by IDT. The oligonucleotide sequences were as follows:

> T7_guideRNA#1_scaffold#1: TTAATACGACTCACTATAGGTTTTCCACGGATCTGCTCGGGTTT TAGAGCTAGAAATAGC
> T7_guideRNA#1_scaffold#2: TTAATACGACTCACTATAGGTGCTTTCTAGGTGAAGCCCCGTTT TAGAGCTAGAAATAGC
> T7_guideRNA#1_scaffold#3: TTAATACGACTCACTATAGGTCATTTGAGCCTAACTACTCGTTT TAGAGCTAGAAATAGC

Along with the above-cited sequences, a reverse primer sequence (AGCACCGACTCGGTGC CACT) from PX335 was used to amplify a PCR-generated template for sgRNA in vitro transcription. The PCR product (~117 bp) was verified on a 2% agarose gel and then gel-purified using the QIAQuick gel extraction kit. The T7 sgRNA PCR product (500 ng) was used as a template for in vitro transcription with the T7 High Yield RNA synthesis kit (NEB, Cat# E2040S). The reaction was incubated at 37°C for 4 hr, and then the sgRNA was purified using the MEGAclear kit (Life Technologies, Cat# AM1908) according to the kit protocol. The sgRNA was eluted with elution buffer preheated to 95°C. The eluted sgRNA was centrifuged at 13,000 rpm for 20 min at 4°C, and the suspension was transferred into a new RNase-free tube. The sgRNA quality was checked by a bioanalyzer using an RNA nano Chip. The sgRNA was diluted to 500 ng/µL in injection buffer (sterile 10 mM Tris/0.1 mM EDTA, pH 7.5) and stored in a −80°C freezer until use. An injection mixture (final volume 30 µL) was prepared in injection buffer by mixing 500 ng/µL of each of the sgRNAs (left and right) with Cas9 mRNA (1 µg/µL, TriLink, Cat# L-6125). Fertilized eggs collected from B6/129 mice were microinjected at the CHOP transgenic core and transferred into pseudo-pregnant B6 females. The pups were genotyped using primers flanking the targeted sequence (forward primer: TTAGGACCTCACCCATCACAA and reverse primer: CATGCCCAGCTACTCT GACAT). The PCR product was cloned into the Promega PGEM T Easy TA cloning vector, and plasmid DNA was Sanger sequenced to determine the size of the deletion. The targeting resulted in mice with a 500 bp and 583 bp deletion at the targeted −83 CNS site and these mutant mice showed same phenotype. The *Il2*-83-cRE mutant heterozygous male mice were backcrossed with B6 females for 10 successive generations.

## ELISA

Cell culture supernatants collected at various time intervals from in vitro stimulated T cells of mice or humans were analyzed for IL-2 by ELISA using kits purchased from Thermo Fisher Scientific (mouse IL-2 ELISA kit: Cat# 88-7024-88) and (human IL-2 ELISA kit, Cat# 88-7025-76). IL-2 ELISA was performed following the protocol provided by the vendor.

## In vivo ovalbumin immunization

Eight-week-old female WT and Il2-CNS-83 KO mice were injected intraperitoneally with 50 µg of ovalbumin (Sigma, Cat# A5503-5G) mixed with IFA (Sigma, Cat# F5506). Blood was collected from the immunized mice after 10 days, and serum was separated. The level of ovalbumin-specific IgG in the serum was determined by ELISA using an ovalbumin-specific IgG ELISA kit purchased from MyBiosource (Cat# MBS763696). The immunized mice were sacrificed on day 10 of immunization, and spleen and lymph nodes were collected. Splenocytes and lymph node cells were stained with CD4 BV785, CD25 BV650, CD44 Percp Cy5.5, CD62L APC eFL780, CXCR5 BV421, PD-1 APC, Bcl-6 PE antibodies. TFH frequency was determined by flow cytometry analysis.

## In vivo inflammatory colitis model

Conventional CD4+CD25-ve T cells and CD4+CD25+Tregs were purified from spleens and lymph nodes of male wild-type or Il2-CNS-83 ko mice (6–8 weeks of age). T cells were transferred into Rag1-ko male mice (five per group) by retro-orbital injection of 1 million wild-type or Il2-CNS-83 ko Tconv cells alone or along with $(0.25 \times 10^6)$ wild-type Tregs. Experimental Rag1-ko recipients were weighed three times per week and scored for IBD-induced clinical symptoms. Mice were sacrificed 72 days post-transfer, and spleen, MLN, and colon were collected. Single-cell suspensions were prepared, cells were stained for CD4, CD8, CD25, CD44, and Foxp3, and analyzed by flow cytometry. The absolute T cell count was estimated using cell count and cell frequency derived from the flow cytometry analysis.

## RNA-seq library generation and sequencing

RNA was isolated from ~1 million of each cell stage using Trizol Reagent (Invitrogen), purified using the Directzol RNA Miniprep Kit (Zymo Research), and depleted of contaminating genomic DNA using DNAse I. Purified RNA was checked for quality on a Bioanlayzer 2100 using the Nano RNA Chip and samples with RIN >7 were used for RNA-seq library preparation. RNA samples were depleted of rRNA using QIAseq Fastselect RNA removal kit (QIAGEN). Samples were then processed for the preparation of libraries using the SMARTer Stranded Total RNA Sample Prep Kit (Takara Bio, USA) according to the manufacturer's instructions. Briefly, the purified first-strand cDNA is amplified into RNA-seq libraries using SeqAmp DNA Polymerase and the Forward and the Reverse PCR Primers from the Illumina Indexing Primer Set HT for Illumina. Quality and quantity of the libraries was assessed using the Agilent 2100 Bioanalyzer system and Qubit fluorometer (Life Technologies). Sequencing was performed on the NovaSeq 6000 platform at the CHOP Center for Spatial and Functional Genomics.

## ATAC-seq library generation and sequencing

A total of 50,000–100,000 sorted cells were centrifuged at $550 \times g$ for 5 min at 4°C. The cell pellet was washed with cold phosphate buffered saline (PBS) and resuspended in 50 µL cold lysis buffer (10 mM Tris–HCl, pH 7.4, 10 mM NaCl, 3 mM $MgCl_2$, 0.1% NP-40/IGEPAL CA-630) and immediately centrifuged at $550 \times g$ for 10 min at 4°C. Nuclei were resuspended in the Nextera transposition reaction mix (25 µL 2× TD Buffer, 2.5 µL Nextera Tn5 transposase (Illumina Cat# FC-121-1030), and 22.5 µL nuclease free $H_2O$) on ice, then incubated for 45 min at 37 °C. The tagmented DNA was then purified using the QIAGEN MinElute kit eluted with 10.5 µL Elution Buffer (EB). 10 µL of purified tagmented DNA was PCR amplified using Nextera primers for 12 cycles to generate each library. PCR reaction was subsequently cleaned up using 1.5× AMPureXP beads (Agencourt), and concentrations were measured by Qubit. Libraries were paired-end sequenced on the Illumina HiSeq 4000 platform (100 bp read length).

## Hi-C library preparation

Hi-C library preparation on FACS-sorted CD4+ T cells was performed using the Arima-HiC kit (Arima Genomics Inc), according to the manufacturer's protocols. Briefly, cells were crosslinked using

formaldehyde. Crosslinked cells were then subject to the Arima-HiC protocol, which utilizes multiple restriction enzymes to digest chromatin. Arima-HiC sequencing libraries were prepared by first shearing purified proximally ligated DNA and then size-selecting 200–600 bp DNA fragments using AmpureXP beads (Beckman Coulter). The size-selected fragments were then enriched using Enrichment Beads (provided in the Arima-HiC kit), and then converted into Illumina-compatible sequencing libraries with the Swift Accel-NGS 2SPlus DNA Library Kit (Swift, 21024) and Swift 2S Indexing Kit (Swift, 26148). The purified, PCR-amplified DNA underwent standard QC (qPCR, Bioanalyzer, and KAPA Library Quantification [Roche, KK4824]) and was sequenced with unique single indexes on the Illumina NovaSeq 6000 Sequencing System using 200 bp reads.

## ATAC-seq data analysis

ATAC-seq peaks from libraries unstimulated and stimulated naïve CD4+ T cells were called using the ENCODE ATAC-seq pipeline (https://www.encodeproject.org/atac-seq/). Briefly, pair-end reads from three biological replicates for each cell type were aligned to hg19 genome using bowtie2 (*Langmead and Salzberg, 2012*), and duplicate reads were removed from the alignment. Narrow peaks were called independently for each replicate using macs2 (*Zhang et al., 2008*) (-p 0.01 `--nomodel --shift` –75 `--extsize` 150 -B --SPMR `--keep-dup` all `--call-summits`) and ENCODE blacklist regions (ENCSR636HFF) were removed from peaks in individual replicates. Reproducible peaks, peaks called in at least two replicates, were used to generate a consensus peakset. Signal peaks were normalized using csaw *Lun and Smyth, 2016* in 10 kb bins background regions and low abundance peaks (counts per million [CPM] > 1) were excluded from the analysis. Tests for differential accessibility were conducted with the glmQLFit approach implemented in edgeR *Robinson et al., 2010* using the normalization factors calculated by csaw. OCR with FDR < 0.05 and abs(log$_2$FC >1) between stages were considered differentially accessible.

## Hi-C data analysis

Paired-end reads from two replicates were preprocessed using the HICUP pipeline v0.7.4 (*Wingett et al., 2015*), with bowtie as aligner and hg19 as the reference genome. The alignment.bam file were parsed to .pairs format using pairtools v0.3.0 (*Abdennur et al., 2023*) and pairix v0.3.7 (*Lee et al., 2022*), and eventually converted to pre-binned Hi-C matrix in .cool format by cooler v0.8.10 (*Abdennur and Mirny, 2020*) with multiple resolutions (500 bp, 1 kbp, 2 kbp, 2.5 kbp, 4 kbp, 5 kbp, 10 kbp, 25 kbp, 40 kbp, 50 kbp, 100 kbp, 250 kbp, 500 kbp, 1 Mbp, and 2.5 Mbp) and normalized with ICE method (*Imakaev et al., 2012*). Replicate similarity was determined using HiCRep v1.12.2 (*Abdennur and Mirny, 2020*) at 10K resolution. For each sample, eigenvectors were determined from an ICE balanced Hi-C matrix with 40 kb resolution using cooltools v0.3.2 (*Nora et al., 2020*) and first principal components were used to determine A/B compartments with GC% of genome region as reference track to determine the sign. Differential TAD comparison was performed using TADcompare with the default settings for each chromosome (v1.4.0) (*Cresswell and Dozmorov, 2020*). Finally, for each cell type, significant intra-chromosomal interaction loops were determined under multiple resolutions (1 kb, 2 kb, and 4 kb) using the Hi-C loop caller Fit-Hi-C2 v2.0.7 (*Kaul et al., 2020*) (FDR < 1e-6) on merged replicates matrix. The consensus chromatin loops within resolution were identified by combining all three stages. These sets of loops were used as consensus for quantitative differential analysis explained below. The final consensus interaction loops for visualization were collected by merging loops from all the resolutions with preference to keep the highest resolution. Quantitative loop differential analysis across cell types was performed on fast lasso normalized interaction frequency (IF) implemented in multiCompareHiC v1.8.0 (*Stansfield et al., 2018*) for each chromosome at resolution 1 kb, 2 kb, and 4 kb independently. The contacts with zero IF among more than 80% of the samples and average IF less than 5 were excluded from differential analysis. The QLF test based on a generalized linear model was performed in cell type-pairwise comparisons, and p-values were corrected with FDR. The final differential loops were identified by overlapping differential IF contacts with consensus interaction loops.

## Bulk RNA-seq data analysis

Bulk RNA-seq libraries were sequenced on an Illumina Novaseq 6000 instrument. The pair-end fastq files were mapped to the genome assembly hg19 by STAR (v2.6.0c) independently for each replicate.

The GencodeV19 annotation was used for gene feature annotation and the raw read count for gene feature was calculated by htseq-count (v0.6.1) (*Anders et al., 2014*) with parameter settings -f bam -r pos -s yes -t exon -m union. The gene features localized on chrM or annotated as rRNAs, small coding RNA, or pseudo genes were removed from the final sample-by-gene read count matrix. Gene set variation analysis was performed using GSVA 1.42.0 (*Hänzelmann et al., 2013*) with the MSigDB hallmark geneset (*Liberzon et al., 2015*), with resulting scores analyzed using limma (limma_3.50.3) (*Ritchie et al., 2015*). Low abundance peaks (CPM > 1) were excluded from the analysis. Testing for differential expression was conducted with the glmQLFit approach implemented in edgeR (*Robinson et al., 2010*). Genes with FDR < 0.05 and abs(log$_2$FC >1) between stages were considered differentially expressed. Differential genes were then clustered using k-means clustering. The number of clusters was determined using the elbow method on the weighted sum of squares, where was set to k = 5. Score for how similar each gene followed the clusters expression pattern was determined by calculating Pearson's correlation coefficients between each gene in the cluster and the cluster centroid.

## Transcription factor footprinting and motif analysis

TF footprints were called using Regulatory Analysis Toolbox HINT-ATAC (v0.13.0) with pooled ATAC-seq data for each stage and consensus peak calls (*Li et al., 2019*). The rgt-hint footprinting was run with parameters –atac-seq, `--paired-end`, and organism =hg19 set. The output footprint coordinates were subsequently matched using rgt-motifanalysis matching with parameters `--organism` hg19 and –pseudocount 0.8 set. The JASPAR2020 position weight matrix database was used to match footprints (*Fornes et al., 2020*). Differential analysis of TF binding across conditions was performed using rgt-hint differential with parameters –organism hg19, `--bc`, `--nc` 24 using the motif matched TF footprints. An activity score is then calculated based on the accessibility surrounding the footprint.

## Partitioned heritability LD score regression enrichment analysis

Partitioned heritability LD score regression (*Finucane et al., 2015*; v1.0.0) was used to identify heritability enrichment with GWAS summary statistics and OCR annotated to genes. The baseline analysis was performed using LDSCORE data (https://data.broadinstitute.org/alkesgroup/LDSCORE) with LD scores, regression weights, and allele frequencies from the 1000G phase 1 data. The summary statistics were obtained from studies as described in *Supplementary file 14* and harmonized with the munge_sumstats.py script. Annotations for partitioned LD score regression were generated using the coordinates of OCR that contact gene promoters through Hi-C loops for each cell type. Finally, partitioned LD scores were compared to baseline LD scores to measure enrichment fold change and enrichment p-values, which were adjusted with FDR across all comparisons.

## V2G mapping using HiC-derived promoter contacts

95% credible sets were determined as previously described (*Su et al., 2021a*). Briefly, p-values from GWAS summary statistics were converted to Bayes factors. Posterior probabilities were then calculated for each variant. The variants were ordered from highest to lowest posterior probabilities added to the credible set until the cumulative sum of posterior probabilities reached 95%. Variants in the 95% credible set were then intersected with the CD4+ T cell promoter interacting region OCR from the three timepoints using the R package GenomicRanges (v1.46.1) (*Lawrence et al., 2013*).

## Genomic reference and visualizations

All analyses were performed using the hg19 reference genome using gencodeV19 as the gene reference. Genomic tracks were visualized with pyGenomeTracks v3.5 (*Lopez-Delisle et al., 2021*). HiC matrices depict the log1p(balanced count) from the cooler count matrix. ATAC-seq tracks were generated from bigwig files that were normalized using deeptools (*Ramírez et al., 2014*).

## ABC model predictions

We used the ABC model (PMID: 31784727) using the our T cell ATAC-seq data (unstimulated or 24 hr stimulated) generated from naive CD4+ T cells merged across replicates. For H3K27ac, we retrieved paired fastq files from H3K27ac MINT-ChIP data from ENCODE for resting (unstimulated) and activated (36 hr stimulated) CD4 T cells derived from thymus. The deduplicated bam files were used as signal files for the ABC model, and the consensus peak set was used as the input set

of enhancers. The accession numbers for unstimulated cells are ENCFF732NOS, ENCFF658TLX, ENCFF605OGN, ENCFF012XYW, ENCFF407QZP, ENCFF556QWM, ENCFF888IHV, ENCFF260FUF, ENCFF550EVW, ENCFF747HGZ, ENCFF067XWQ, ENCFF939YBH, ENCFF749LGW, ENCF-F788LWV, ENCFF610FRB, ENCFF538GQX, ENCFF355PCA, ENCFF040OMT, ENCFF878SHZ, ENCFF210WNQ, ENCFF849ZJH, ENCFF925ARM, ENCFF112YNU, ENCFF808JFV, ENCFF448KHE, ENCFF591BFV, ENCFF366UTF, and ENCFF172EWR. The accession numbers for stimulated cells are ENCFF442DFF, ENCFF870ZYC, ENCFF189UPS, ENCFF725KSA, ENCFF618YAV, ENCFF765AJK, ENCFF854QUS, ENCFF180TKV, ENCFF761RCK, ENCFF449EAE, ENCFF915VPK, ENCFF575XFH, ENCFF227AGX, ENCFF810XFH, ENCFF367OMH, ENCFF114BBZ, ENCFF855XPP, ENCFF132AKN, ENCFF663MHJ, ENCFF244TPB, ENCFF230RIJ, ENCFF648FPC, ENCFF797CIK, ENCFF096JPW, ENCFF704KMT, ENCFF385POD, ENCFF395RIG, ENCFF043RRJ, ENCFF027RNR, ENCF-F333RGD, ENCFF048PMV, ENCFF887XSA, ENCFF188CNA, ENCFF614IZY, ENCFF984TTZ, ENCF-F977RDC, ENCFF393EWO, ENCFF781VQT, ENCFF638UCZ, ENCFF873QZH, ENCFF553JPM, ENCFF589HRL, ENCFF721ZCN, ENCFF496HYV, ENCFF455WBD, ENCFF679UTJ, ENCFF200MYN, and ENCFF459ARH.

The retrieved files were trimmed/preprocessed using fastp with the default settings. Following this, matched R1 and R2 files were concatenated to a single R1/R2 file pair. The reads were aligned to hg19 and duplicates were removed as described for the ATAC-seq data analysis. The deduplicated bam files were used as input to the ABC model. We ran the ABC model with the default settings for hg19 and provided reference files for the blacklist, promoter annotation, chromosome sizes reference, and set of ubiquitously expressed genes. The HiC data used was the KR normalized averaged HiC dataset that was derived from a set of cell lines (GM12878, NHEK, HMEC, RPE1, THP1, IMR90, HU- VEC, HCT116, K562, KBM7; average_hic.v2.191020.tar.gz) at 5 kb resolution, then scaled by powerlaw distribution as described in the original manuscript. The default threshold of 0.2 was used to link enhancers to genes. The resulting elements were intersected with the credible set list used in this study for precision-recall analyses.

## Precision-recall analyses against HIEI genes

To benchmark our autoimmune effector gene predictions against prior eQTL and chromatin capture studies, we curated a list of 449 expert curated genes from Human Errors in Inborn Immunity 2022, where mutations have been reported to cause immune phenotypes in humans (*Tangye et al., 2022*). Most of these genes were identified through rare loss-of-function mutations in the coding region. In addition to this, we also took a subset of 145 genes reported to specifically result in autoimmune phenotypes. These two sets of genes were treated as 'gold standard' sets of genes to compare different approaches to predict GWAS effector genes. We curated the results of several other chromatin conformation or eQTL-based methods of assigning variants to genes to compare with our study. For the chromatin confirmation studies, we obtained loop calls from the associated publications and intersected with the list variants in the 95% credible set from 15 GWAS used to predict effector genes using GenomicRanges (v1.46.1) (*Lawrence et al., 2013*). For eQTL studies, we obtained the summary stats files and identified eGenes linked to any member of the credible set reported as an eQTL in the transcriptome-wide association analysis (TWAS; p-value<1e-4). Precision was calculated as the ratio of the number predicted genes in the truth set to the total number of predicted genes, while recall was calculated as the ratio of the number predicted genes in the truth set to the total number of genes in truth set.

## Co-localized eQTL comparisons

In addition to comparisons with the eGenes identified by TWAS in the *Soskic et al., 2022* study, we also compared overlap of gene nominations in our study with the eQTLs that co-localized with an autoimmune GWAS. Enrichment of sentinel-gene assignments was conducted similarly as described previously (*Su et al., 2020*). Briefly, a null distribution was constructed by randomly selecting genes within 1 Mb of the sentinel compared to the set of co-localized *cis*-eQTL *Soskic et al., 2022* found in through 10,000 iterations. The observed overlap reports the set of gene identified by both our HiC-based approach with the set of co-localized eQTLs. We report the empirical p-value of the observed value relative to null distribution.

## International mouse phenotyping consortium comparisons

The set of HiC implicated genes were compared to the mouse international phenotyping consortium set of genes with reported phenotypes (*Muñoz-Fuentes et al., 2018*). We converted the list of V2G implicated genes to mouse homologs using homologene. We tested for enrichment for each phenotype using a one-sided proportion test implicated in R prop.test with type set to 'upper'.

## Identification of pharmacological agents

We queried the Drug-Gene Interaction Database with the set of V2G implicated genes for chemicals using rDGIdb (v1.20.0) (*Wagner et al., 2016*). To identify the number of papers for each gene with at least one drug annotated to target it, we queried PubMed titles and abstracts using the R package RISmed (v2.3.0) with each gene's name and either 'autoimmune' and the list of autoimmune diseases (*Supplementary file 17*). A score to approximate expression specificity was computed using the sum GTEX median expression values (v8) for whole blood or spleen divided by other tissues (*GTEx Consortium, 2015*).

## Lentiviral-based CRISPR/CAS9 targeting in Jurkat cells

LentiCRISPRv2-mCherry vectors encoding gRNA-CAS9 and the fluorescent reporter mCherry were used for Jurkat targeting. CRISPR guide RNAs (sgRNA) targeting human IL-2-21 intergenic -46, -51, -80, -83, -85, -122, -128 CNS regions were designed using http://crispr.tefor.net and cloned into lentiCRISPRv2-mCherry. Empty vector without gRNA insert was used as a control. Below is the list of CRISPR gRNA for causing deletion of human IL2-21 intergenic regions:

| CNS region | CRISPR gRNA | |
|---|---|---|
| –46 CNS | gRNA #1 | AGGATGCCTACCTCCAAATG |
| | gRNA # 2 | AGGTGACAACATTTAGTCAG |
| –51 CNS | gRNA #1 | GGCAACGAAATTCACTGTGA |
| | gRNA # 2 | ATTCTAACAGGAATCATTCG |
| –80 CNS | gRNA # 1 | GTTCTACCTATGCCGCATTG |
| | gRNA # 2 | GAGATTTACTCAGTCCAATG |
| –83 CNS | gRNA # 1 | GTGACAAGCATGACTCTACA |
| | gRNA # 2 | GTGATGGTGAATTAAGCTGA |
| –85 CNS | gRNA #1 | AGGGTTTTCTAGTTACGAGA |
| | gRNA # 2 | ATGGTTAGTTAGCTCCCAAG |
| –96 CNS | gRNA#1 | TGGGAAAAACATCTTACCTG |
| | gRNA # 2 | TGGCCCATGAACCATCAAAG |
| –122 CNS | gRNA# 1 | GTTATTAATCTAAGCGGAGA |
| | gRNA# 2 | GGAAGTTAGGCAGTCAATCG |
| –128 CNS | gRNA#1 | CTTCAATCATTGCATTCCAC |
| | gRNA# 2 | TGACACCACCCCTGCTTGAG |

HEK 293T cells were grown in RPMI1640 complete medium (RPMI1640 + 1× P/S, 1× L-Glu, 10% FBS[fetal bovine serum]), 37°C, 7% $CO_2$. 293T cells were transfected with 10 ug of lenti-CRISPR-V2-CRE construct along with packaging plasmid 6 ug of PsPAX2 (Addgene, Cat# 12260) and 3.5 ug of PmD2.G (Addgene, Cat# 12259) using Lipofectamine 2000 transfection reagent (Invitrogen Cat# 11668019). After 6 hr, the transfection medium was replaced with complete culture medium. Transfected cells were incubated at 37°C for 48–72 hr in a cell culture incubator. and then the Lentiviral supernatants were harvested and spun at 300 × *g* for 5 min to remove cellular debris. Lentiviral supernatants were concentrated using Lenti-X Concentrator (Takara Bio, Cat# 631232) and then centrifuged at 1500 × *g* for 30 min at 4°C and supernatant was discarded. The lentiviral pellet was resuspended at a ratio of 1:20 of the original volume using RPMI media, and concentrated virus supernatant aliquots

were prepared and stored until use at –80°C. To achieve high transduction efficiency, the viral supernatant was titrated in Jurkat cells through transduction using various dilutions of the viral supernatants and transduction efficiency was determined by mCherry expression analyzed through flow cytometry. Jurkat cells were seeded in a 24-well plate at $0.5 \times 10^6$/well in culture media, and viral supernatant with 8 ug/mL of polybrene was added to each well. Spinfection was performed for 90 min at 2500 rpm, and transduced cells were equilibrated at 37°C for ~6 hr, followed by incubation at 37°C 5% $CO_2$ for ~72 hr culture. Transduced Jurkat cells were then harvested and stimulated using PMA (15 ng/mL) + ionomycin (1 uM) + human anti-CD28 (2 ug/mL), BioXcell Cat# BE0248 in 96-well culture plates (TPP, Cat# 92097) in triplicates. Cell culture supernatants were collected at the end of culture and analyzed for IL-2 by ELISA using a kit (Cat# 88-7025-76) purchased from Thermo Fisher Scientific.

## gRNA-CAS9 RNP-based targeting in primary human CD4+ T cells

Primary human CD4 T cells derived from five normal healthy donors were obtained from Human Immunology core (University of Pennsylvania). Alt-R S.p. HiFi Cas9 Nuclease V3 Cat# 1081061 CAS9 and following list of Alt-R CRISPR-Cas9 sgRNA targeting IL-2–21 CRE were purchased from Integrated DNA Technologies, USA.

| Name | 20 nt gRNA seq |
| --- | --- |
| 46CNS-1 | GACGTATATAGTCATCTGAT |
| 46CNS-2 | TCTGTGGAGCTGCTGCGTTA |
| 46CNS-3 | AGGATGCCTACCTCCAAATG |
| 51CNS-1 | ATTCTAACAGGAATCATTCG |
| 51CNS-2 | GAGTAAAAAGACGTGTTACC |
| 51CNS-3 | GGCAACGAAATTCACTGTGA |
| 80CNS-1 | AATGCGGCATAGGTAGAACT |
| 80CNS-2 | GAGATTTACTCAGTCCAATG |
| 80CNS-3 | TGGTTGTCACAGTAACTAGG |
| 83CNS-1 | GTGACAAGCATGACTCTACA |
| 83CNS-2 | GTGATGGTGAATTAAGCTGA |
| 83CNS-3 | TGGGCTCTGACTCACTTAGA |
| 85CNS-1 | GCTAACATTGACTTCTCTAC |
| 85CNS-2 | AATCTATGCAAGGGGTGAAT |
| 85CNS-3 | GCATCATGATGAAGCTTATC |
| 122CNS-1 | GTTATTAATCTAAGCGGAGA |
| 122CNS-2 | GGAAGTTAGGCAGTCAATCG |
| 122CNS-3 | CACTTTGTGCTCGGATGCTC |
| 128CNS-1 | CTTCAATCATTGCATTCCAC |
| 128CNS-2 | TCAAGCAGGGGTGGTGTCAA |
| 128CNS-3 | TGGTGATTCATCTTTAGCAT |
| IL-2URR-1 | TCCATTCAGTCAGTCTTTGG |
| IL-2URR-2 | TAGTGTCCCAGGTGATTTAG |
| IL-2URR-3 | TAGAGCTATCACCTAAGTGT |
| B2m-1 | GAGTAGCGCGAGCACAGCTA |
| B2m-2 | CACGCGTTTAATATAAGTGG |

Primary human CD4 T (5-10e6) were incubated with sgRNA and CAS9 protein complex, and electroporation was done using P3 Primary Cell 4D-Nucleofector X Kit L (Lonza, Cat# V4XP-3024) and

Lonza 4D nucleofector system. B2M gene was CRISPR targeted as a positive control. As per the manufacturer's protocol, cells were electroporated pulse code Fl-115 in 100 uL cuvette format. After nucleofection, cells were allowed to rest in the complete media for ~2 days. Cells were then harvested, washed with PBS, and aliquots of cells were used for further experimentation such as flow staining and cell activation. Primary human CD4 T cells (0.1e6/well) were seeded in triplicates (for each experimental condition) in 96-well plate format in RPMI complete medium and stimulated using ImmunoCult Human CD3/CD28 T Cell Activator (STEM Cells Technologies, Cat# 10971). Cell culture supernatants were collected at 4 hr of stimulation and stored at –80°C until assayed. IL-2 ELISA was performed using Thermo Fisher Scientific IL-2 Human Uncoated ELISA Kit with Plates, Cat# 88-7025-76.

## Recombinant reporter assays

The *IL2* URR was cloned into XhoI and HindIII sites of PGL4 Luc vector (Promega) and then ~500 bp individual sequence representing *IL2-IL21* intergenic CNS at −46,−51, −80,−83, −85, −128 were cloned upstream to IL-2 URR at the XhoI site of the pGL4 vector. Primary human CD4 T cells obtained from five normal healthy donors were transfected with PGL4-cRE-URR constructs. Briefly, primary human CD4 T cells were activated with anti-CD3+anti-CD28 Dynabeads (1:1) (Thermo Fisher scientific, Cat# 11161D)+IL-2 overnight, and then 1 million aliquots of cells (triplicates) were electroporated using Nucleofector 2b, human T cell Nuclefactor kit (Lonza VPA# 1002, program# T-020) with 2 ug of PGL4 firefly vector constructs along with 0.2 ug of PGL4 Renilla vector; cells were allowed to rest overnight in RPMI+IL-2 and then re-stimulated with plate-bound anti-CD3+anti-CD28 (2 ug/mL each) for 5 hr. Dual luciferase assay was performed with cell lysate prepared using Promega dual luciferase assay kit. Cell lysate was prepared in PLB as per the manufacturer's protocols, and then firefly and renilla luciferase activities were analyzed using spectramax ID5 (Molecular Devices). Firefly luciferase activity was normalized against the internal control renilla luciferase activity.

## Pharmacological validation of novel T cell activation regulatory genes

The drugs lometrexol (GART antagonist), 25-hydroxycholesterol (GPR183 ligand), epinephrine and norepinephrine (CHRNB agonists), and arachidonoyl glycine (GPR18 agonist) were purchased from Sigma. The GPR18 antagonist O-1918 was purchased from ChemCruz. CEP1347 (MAP3K11 antagonist), kisspeptin 234 (KISSIR ligand) and the PTK6 antagonist tilfrinib were purchased from Tocris. The GRK6 antagonist GRK-IN-1 was purchased from DC Chemicals. The SIK1 antagonist HG-91-9-1 was purchased from Selleckchem. The OXER1 agonist 5-Oxo-ete was purchased from Cayman Chemicals. The drugs were dissolved in DMSO or ethanol as suggested by the vendor, and working stocking concentrations were prepared in RPMI1640 medium or PBS. The drug effect on T cell proliferation was assayed using murine lymphocytes cultured under TCR and CD28 activation conditions. Spleen and lymph nodes were collected from 6- to 8-week-old female C57BL/7 mice, and single-cell lymphocyte cell suspensions were prepared in RPMI 1640 complete medium. 20 million lymphocytes were labeled with CFSE and re-suspended in RPMI 1640 medium. Cells (0.5 million labeled cells/well) were loaded into 48-well culture plate and activated with mouse anti-CD3 and anti-CD28 agonistic antibodies (1 μg mL each). Drugs at the indicated concentrations were added in the culture medium, and both untreated and drug treated cell cultures were incubated at 37°C for 72 hr in a cell culture incubator. Cells were harvested after 3 days of culture, washed with PBS, and then stained with live-dead aqua dye. After washing with FACS buffer (PBS containing 2% FBS), cells were stained with fluorochrome conjugated antibodies CD4-APC, CD8-PB, CD44-Percp Cy5.5, and CD25-BV650. Stained cells were analyzed on a Beckman Coulter Cytoflex S flow cytometer. The division profile of CD4+CFSE+ T cells was gated on live populations. The flow data was analyzed using Flowjo10 software, and the number of divided CD4+ T cells was determined as described previously (*Wells et al., 1997*).

## Additional information

### Funding

| Funder | Grant reference number | Author |
|---|---|---|
| National Institutes of Health | R01AI137143 | Daniel Campbell<br>Andrew D Wells |
| National Institutes of Health | R01DK122586 | Struan FA Grant<br>Andrew D Wells |
| National Institutes of Health | R21AI110179 | Andrew D Wells |

The funders had no role in study design, data collection and interpretation, or the decision to submit the work for publication.

### Author contributions

Matthew C Pahl, Conceptualization, Resources, Data curation, Software, Formal analysis, Investigation, Visualization, Methodology, Writing - original draft; Prabhat Sharma, Data curation, Formal analysis, Validation, Investigation, Methodology; Rajan M Thomas, Zachary Thompson, Zachary Mount, James A Pippin, Peng Sun, Validation, Investigation, Methodology; Peter A Morawski, Investigation, Methodology; Chun Su, Data curation, Formal analysis; Daniel Campbell, Conceptualization, Supervision, Funding acquisition; Struan FA Grant, Conceptualization, Supervision, Funding acquisition, Project administration; Andrew D Wells, Conceptualization, Resources, Supervision, Funding acquisition, Investigation, Project administration, Writing - review and editing

### Author ORCIDs

Matthew C Pahl http://orcid.org/0000-0002-4720-1921
Struan FA Grant https://orcid.org/0000-0003-2025-5302
Andrew D Wells https://orcid.org/0000-0002-3630-2145

### Ethics

This study was performed in accordance with the recommendations in the Guide of the Care and Use of Laboratory Animals of the National Institutes of Health at the AAALAC-accredited Children's Hospital of Philadelphia vivarium under an IACUC-approved protocol. IACUC-approved protocol 22-000594.

Reviewer #1 (Public Review): https://doi.org/10.7554/eLife.96852.3.sa1
Reviewer #2 (Public Review): https://doi.org/10.7554/eLife.96852.3.sa2
Reviewer #3 (Public Review): https://doi.org/10.7554/eLife.96852.3.sa3
Author response https://doi.org/10.7554/eLife.96852.3.sa4

# Additional files

### Supplementary files

- Supplementary file 1. Expression and clustering of differentially expressed genes.
- Supplementary file 2. Pathway enrichment of differentially expressed RNA-seq genes.
- Supplementary file 3. Pathway enrichment of differentially expressed RNA-seq genes by cluster.
- Supplementary file 4. Accessible chromatin regions.
- Supplementary file 5. Differential accessible open chromatin regions.
- Supplementary file 6. A/B compartment calls.
- Supplementary file 7. Differential TAD boundaries.
- Supplementary file 8. Stripe calls.
- Supplementary file 9. Differential contact frequency.
- Supplementary file 10. Summary of TF footprinting.
- Supplementary file 11. TF target gene pathway enrichment.

- Supplementary file 12. List of all GWAS included.
- Supplementary file 13. Variant to gene mapping across all timepoints.
- Supplementary file 14. Partitioned LD score regression output.
- Supplementary file 15. Autoimmune variants in open gene promoters.
- Supplementary file 16. Comparison of V2G approaches.
- Supplementary file 17. Motifs predicted to disrupt transcription factor binding sites.
- Supplementary file 18. V2G genes implicated by orthogonal data.
- Supplementary file 19. V2G target gene drug repurposing results.
- MDAR checklist

## Data availability

Sequencing data of been deposited in GEO under accession GSE230346. Mutant mouse strains created as part of this work will be made available after publication for non-commercial research by written request. Individual requests for shipment of mice generated by this project to the Association for Assessment and Accreditation of Laboratory Animal Care International (AAALAC)-accredited institutions will be honored. The recipient investigators would provide written assurance and evidence that the animals will be used solely in accord with their local IACAC review, that animals will not be further distributed by the recipient without written consent, and that the animals will not be used for commercial purposes.

The following dataset was generated:

| Author(s) | Year | Dataset title | Dataset URL | Database and Identifier |
|---|---|---|---|---|
| Pahl M, Grant S, Wells A | 2024 | Chromatin conformation dynamics during CD4+ T cell activation implicates 2 autoimmune disease-associated genes and regulatory elements | https://www.ncbi.nlm.nih.gov/geo/query/acc.cgi?acc=GSE230346 | NCBI Gene Expression Omnibus, GSE230346 |

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
