## [Editor Report · eLife assessment]

This is a **solid** study that follows a well-established canvas for variant-to-gene prioritization using 3D genomics, applying it to activated T cells. The authors go some way in validating the lists of candidate genes, as well as exploring the regulatory architecture of a candidate GWAS locus. Jointly with data from previous studies performing variant-to-gene assignment in activated CD4 T cells (and other immune cells), this work provides a **useful** additional resource for interpreting autoimmune disease-associated genetic variation.

---

## [Referee Report · Reviewer #1 (Public Review)]

Summary:

The authors profile gene expression, chromatin accessibility and chromosomal architecture (by Hi-C) in activated CD4 T cells and use this information to link non-coding variants associated with autoimmune diseases with putative target genes. They find over a 1000 genes physically linked with autoimmune disease loci in these cells, many of which are upregulated upon T cell activation. Focusing on IL2, they dissect the regulatory architecture of this locus, including the allelic effects of GWAS variants. They also intersect their variant-to-gene lists with data from CRISPR screens for genes involved in CD4 T cell activation and expression of inflammatory genes, finding enrichments for regulators. Finally, they showed that pharmacological inhibition of some of these genes impacts T cell activation.

This is a solid study that follows a well-established canvas for variant-to-gene prioritisation using 3D genomics, applying it to activated T cells. The authors go some way in validating the lists of candidate genes, as well as explore the regulatory architecture of a candidate GWAS locus. Jointly with data from previous studies performing variant-to-gene assignment in activated CD4 T cells (and other immune cells), this work provides a useful additional resource for interpreting autoimmune disease-associated genetic variation.

Autoimmune disease variants were already linked with genes in CD28-stimulated CD4 T cells using chromosome conformation capture, specifically Promoter CHi-C and the COGS pipeline (Javierre et al., Cell 2016; Burren et al., Genome Biol 2017; Yang et al., Nat Comms 2020). The authors cite these papers and present a comparative analysis of their variant-to-gene assignments (in addition to scRNA-seq eQTL-based assignments). Furthermore, they find that the Burren analysis yields a higher enrichment for gold standard genes.

I thank the authors for their revisions in response to my initial review. The revised version now includes a more comprehensive comparative analysis of different datasets and V2G approaches and discusses the potential sources of differences in the results. Most significantly, the authors have now included an interesting comparison of their methodology with the popular ABC technique and outlined the key limitations of ABC relative to their method and other (Capture) Hi-C-based V2G approaches.

---

## [Referee Report · Reviewer #2 (Public Review)]

Summary:

There is significant interest in characterizing the mechanisms by which genetic mutations linked to autoimmunity perturb immune processes. Pahl et al. collect information of dynamic accessible regions, genes, and 3D contacts in primary CD4+ T cell samples that have been stimulated ex vivo. The study includes a variety of analyses characterizing these dynamic changes. With TF footprinting they propose factors linked to active regulatory elements. They compare the performance of their variant mapping pipeline that uses their data versus existing datasets. Most compelling there was a deep dive into additional study of regulatory elements nearby the IL2 gene. Finally, they perform a pharmacological screen targeting several genes they suggest are involved in T cell proliferation.

Strengths:

- The work done characterizing elements at the IL2 locus is impressive.

Weaknesses:

- There are extensive studies performed on resting and activated immune cell states (CD4+ T cells and other cell types) and some at multiple time points or concentrations of stimuli that collect ATAC-seq and/or RNA-seq. Several analyses performed in published studies were similarly performed in this study. I expected the authors to at least briefly mention published studies and whether their conclusions generally agree or disagree. Are the same dynamic regulatory regions or genes identified upon T cell activation? Are the same TF footprints enriched in these dynamic regulatory elements? In the revision, I appreciate that the authors now include additional data from several studies that I had initially suggested for the purposes of nominating disease genes in their precision-recall analysis.

---

## [Referee Report · Reviewer #3 (Public Review)]

Summary:

This paper used RNAseq, ATACseq, and Hi-C to assess gene expression, chromatin accessibility, and chromatin physical associations for native CD4+ T cells as they respond to stimulation through TCR and CD28. With these data in hand, the author identified 423 GWAS signals to their respective target genes, where most of these were not in the proximal promoter, but rather distal enhancers. The IL-2 gene was used as an example to identify new distal cis regulatory regions required for optimal IL-2 gene transcription. These distal elements interact with the proximal IL2 promoter region. When the distal enhancer contained an autoimmune SNP, it affected IL-2 gene transcription. The authors also identified genetic risk variants that were associated to genes upon activation. Some of these regulate proliferation and cytokine production, but others were novel.

Strengths:

This paper provides a wealth of data related to gene expression after CD4 T cells are activated through the TCR and CD28. An important strength of this paper is that these data were intensively analyzed to uncover autoimmune disease SNPs in cis acting regions. Many of these could be assigned to likely target genes even though they often are in distal enhancers. These findings help to provide a better understanding concerning the mechanism by which GWAS risk elements impact gene expression.

Another strength to this study was the proof-of-principle studies examining the IL-2 gene. Not only were new cis acting enhancers discovered, but they were functionally shown to be important in regulating IL-2 expression, including susceptibility to colitis. Their importance was also established with respect to such distal enhancers harboring disease relevant SNPs, which were shown to affect IL-2 transcription.

The data from this study were also mined against past Crispr screens that identified genes that control aspects of CD4 T cell activation. From these comparisons, novel genes were identified that function during T cell activation.

Weaknesses:

A weakness from this study is that few individuals were analyzed, i.e., RNAseq and ATACseq (n = 3) and HiC (n = 2). Thus, the authors may have underestimated potentially relevant risk associations by their chromatin capture-based methodology. This might account for low overlap of their data with the eQTL-based approach or the HIEI truth set.

The authors explain that the low overlap is not due to few GWAS associations by HiC. The expanded discussion in the revised manuscript provides a framework to help explain inherent differences between these methods that may contribute to the low overlap.

Impact:

This study indicates that defining distal chromatin interacting regions help to identify distal genetic elements, including relevant variants, that contribute to gene activation.

---

## [Author Response]

The following is the authors’ response to the original reviews.

**Reviewer #1 (Public Review):**
Summary:The authors profile gene expression, chromatin accessibility, and chromosomal architecture (by Hi-C) in activated CD4 T cells and use this information to link non-coding variants associated with autoimmune diseases with putative target genes. They find over 1000 genes physically linked with autoimmune disease loci in these cells, many of which are upregulated upon T cell activation. Focusing on IL2, they dissect the regulatory architecture of this locus, including the allelic effects of GWAS variants. They also intersect their variant-to-gene lists with data from CRISPR screens for genes involved in CD4 T cell activation and expression of inflammatory genes, finding enrichments for regulators. Finally, they showed that pharmacological inhibition of some of these genes impacts T-cell activation.This is a solid study that follows a well-established canvas for variant-to-gene prioritisation using 3D genomics, applying it to activated T cells. The authors go some way in validating the lists of candidate genes, as well as exploring the regulatory architecture of a candidate GWAS locus. Jointly with data from previous studies performing variant-to-gene assignment in activated CD4 T cells (and other immune cells), this work provides a useful additional resource for interpreting autoimmune disease-associated genetic variation.Suggestions for improvement:Autoimmune disease variants were already linked with genes in CD28-stimulated CD4 T cells using chromosome conformation capture, specifically Promoter CHi-C and the COGS pipeline (Javierre et al., Cell 2016; Burren et al., Genome Biol 2017; Yang et al., Nat Comms 2020). The authors cite these papers and present a comparative analysis of their variant-to-gene assignments (in addition to scRNA-seq eQTL-based assignments). Furthermore, they find that the Burren analysis yields a higher enrichment for gold standard genes.The obvious question that the authors don't venture into is why the results are quite different. In principle, this could be due to the differences between:(a) the cell stimulation procedure(b) the GWAS datasets used(c) the types of assay (Hi-C vs Capture Hi-C)(d) approaches for defining gene-linked regions (loops vs neighbourhoods)(e) how the GWAS signals at gene-linked regions are aggregated (e.g., the flavours of COGS in Javierre and Burren vs the authors' approach)Re (a), I'm not sure the authors make it explicitly clear in the main text that the Capture Hi-Cbased studies also use *stimulated* CD4 T cells, particularly in the section "Comparative predictive power...". So the cells used are pretty much the same, and the differences likely arise from points (b) to (e).It would be useful for the community to understand more clearly what is driving these differences, ideally with some added data. Could the authors, for example, take the PCHi-C data from Javierre/Burren and use their GWAS data and variant-to-gene assignment algorithms?

We greatly appreciate the referee’s expert assessment of our work and its value to the field, and we are glad that the referee was enthused by our comparison of the predictive power of the various V2G approaches. A point not emphasized enough in the original version of the manuscript is that we actually did harmonize the various datasets in the way the referee suggests for the precision/recall analysis. We took the contact maps presented from each paper, mapped genes using the same set of GWAS SNPs, and defined all gene-linked regions using our loop calling approach. This has been clarified in the revised version of the manuscript. We have now included a more thoughtful discussion of the possible sources of discrepancy between the different studies included in the comparison, and our thoughts on the potential sources raised by the referee are outlined below:

(a) The modes of stimulation used are similar between studies, but timepoints and donors did vary, and ours was the only study that sorted naïve CD4+ T cells before stimulation. These aspects could represent a source of variability.

(b) The GWAS is not a source of variability because we re-ran the raw data from all the orthogonal studies through our V2G pipeline using the same GWAS as in the current manuscript.

(c) The use of HiC vs. Capture HiC is a likely source of variability. The Capture-HiC datasets included in our comparison are lower resolution (i.e. HindIII) but focus higher sequencing depth at promoters compared to our HiC datasets – i.e., Capture-HiC may mis-call loops to the wrong promoters due to lower resolution as we have shown in our previous study [Su, Human Genetics, 2021], and will miss distal SNP interactions at promoters not included in the capture set. While HiC is unbiased in this regard, HiC will fail to call some SNP-promoter loops called by CaptureHiC because the sequencing power is not specifically focused at promoters.

(d) For studies using neighborhood approaches, we re-ran the raw data through our loop calling algorithm to connect distal SNP to gene promoters, and regarding (e) above, we ran the raw data through our V2G pipeline to allow a better comparison.

In addition, given that the authors use Hi-C, a popular method for V2G prioritisation for this type of data is currently ABC (Nasser et al, Nature 2021). Could the authors provide a comparative analysis with respect to the V2G assignments in the paper and, if they see it appropriate, also run ABC-based GWAS integration on their own Hi-C data?

This is an excellent suggestion, which we have followed in the revised version of our manuscript. It should be noted (and we do so in the text of the revision) that there is an important caveat to bringing in the ABC model. Chromosome conformation-based approaches are biologically constrained (i.e., informed) by the natural structure of chromatin in the nucleus that controls how gene transcription is regulated in cis, and it does so in a way that brings value to GWAS data. However, the ABC model further constrains the input data by imposing non-biological filters that allow the algorithm to be applied, but impose artifactual limitations that may negatively impact interpretation and discovery. In addition to filtering out pseudogenes, bidirectional RNA, antisense RNAs, and small RNAs, the ABC model gene set eliminates genes ubiquitously expressed across tissues (based on the assumption that these genes are driven primarily by elements adjacent to their promoters) and only allows annotation of one promoter per gene, even though the median number of promoters per gene in the human genome is three. In contrast, our chromatin-based V2G removes pseudogenes, but includes lincRNA and small RNAs, and includes all alternative transcription start sites annotated by gencode.

To apply the ABC GWAS gene nomination model to our CD4+ T cell chromatin-based V2G data, we used our ATAC-seq data and publicly available CD4+ T cell H3K27ac ChIP-seq data as input, and integrated this with GWAS and the average ENCODE-derived HiC dataset from the original ABC paper. The activity-by-contact model nominated 650 genes, compared to 1836 genes when using our cell type-matched HiC data and analysis pipeline. Only 357 of these genes were nominated by both approaches; 1479 genes nominated by our approach were not nominated by ABC, while 293 genes not implicated by our approach were newly implicated by ABC. To determine how the ABC-constrained approach performs against the HIEI gold standard set, we subjected all datasets used for the comparison depicted in the new Figure 5D to the same promoter filter used by the ABC model prior as part of the precision-recall re-analysis. Firstly, we found that applying the restricted ABC model promoter annotation to all datasets did not have a large effect on recall, however, the precision of several of the datasets were affected. For example, using the restricted promoter set reduced the precision of our (Pahl) V2G approach and inflated the precision of the nearest gene to SNP metric. Second, the new precision-recall analysis shows that the ABC score-based approach is only half as sensitive at predicting HIEI genes as the chromatin-based V2G approaches. This indicates that constraining GWAS data with cell type- and state-specific 3D chromatin-based data brings more GWAS target gene predictive power than application of the multi-tissue-averaged HiC used by the ABC model. We thank the reviewer for helpful suggestions that have improved the quality of our study.

**Reviewer #2 (Public Review):**
Summary:There is significant interest in characterizing the mechanisms by which genetic mutations linked to autoimmunity perturb immune processes. Pahl et al. collect information on dynamic accessible regions, genes, and 3D contacts in primary CD4+ T cell samples that have been stimulated ex vivo. The study includes a variety of analyses characterizing these dynamic changes. With TF footprinting they propose factors linked to active regulatory elements. They compare the performance of their variant mapping pipeline that uses their data versus existing datasets. Most compelling there was a deep dive into additional study of regulatory elements nearby the IL2 gene. Finally, they perform a pharmacological screen targeting several genes they suggest are involved in T cell proliferation.Strengths:The work done characterizing elements at the IL2 locus is impressive.Weaknesses:Missing critical context to evaluate claims. There are extensive studies performed on resting and activated immune cell states (CD4+ T cells and other cell types) and some at multiple time points or concentrations of stimuli that collect ATAC-seq and/or RNA-seq that have been ignored by this study. How do conclusions from previous studies compare to what the authors conclude here? It is impossible to evaluate the claims without this additional context. These are a few studies I am familiar with (the authors should perform a more comprehensive search to be sure they're not ignoring existing observations) that would be important to compare/contrast conclusions: o Alasoo, K. et al. Shared genetic effects on chromatin and gene expression indicate a role for enhancer priming in immune response. Nat. Genet. 50, 424-431 (2018).- Calderon, D., Nguyen, M.L.T., Mezger, A. et al. Landscape of stimulation-responsive chromatin across diverse human immune cells. Nat Genet 51, 1494-1505 (2019).- Gate, R.E., Cheng, C.S., Aiden, A.P. et al. Genetic determinants of co-accessible chromatin regions in activated T cells across humans. Nat Genet 50, 1140-1150 (2018). o Glinos, D.A., Soskic, B., Williams, C. et al. Genomic profiling of T-cell activation suggests increased sensitivity of memory T cells to CD28 costimulation. Genes Immun 21, 390-408 (2020). o Gutierrez-Arcelus, M., Baglaenko, Y., Arora, J. et al. Allele-specific expression changes dynamically during T cell activation in HLA and other autoimmune loci. Nat Genet 52, 247-253 (2020).- Kim-Hellmuth, S. et al. Genetic regulatory effects modified by immune activation contribute to autoimmune disease associations. Nat. Commun. 8, 266 (2017). o Ye, C. J. et al. Intersection of population variation and autoimmunity genetics in human T cell activation. Science 345, 1254665 (2014).- As a general point, I appreciate it when each claim includes a corresponding effect size and p-value, which helps me evaluate the strength of significance of supporting evidence.

We greatly appreciate the referee’s expert assessment of our work and emphasis on the value of our functional follow-up studies. Our precision-recall analyses were not meant to represent an exhaustive comparison of all prior GWAS gene nomination studies, although we agree that this could (and should) be done as part of a separate study in a future manuscript. Instead, we focused on gene nomination studies that (1) analyzed resting and activated human CD4+ T cells, (2) whose experimental design was most comparable to our own studies, and (3) had raw data readily available in the appropriate formats to allow re-analysis and harmonization before comparison. This is a point we did not make sufficiently clear in the original version of the manuscript, but have clarified in the revision.

Based on this rationale, we agree that the studies by Gate et al. and Ye et al. should be included in our comparative precision-recall analysis, and we have done so in the revised manuscript. The Gate study reported ATAC-seq peak co-accessibility, caQTL, eQTL, and HiC data, and we now include the resulting gene nominations from these datasets in the precision-recall analysis. These datasets performed poorly with respect to nomination of HIEI genes, likely due to small sample numbers and low sequencing depth compared to the other eQTL and chromatin capture-based studies. The eQTL reported by Ye et al. nominated 15 genes for autoimmune traits, two of which were in the ‘truth’ HIEI set (IL7R and IL2RB). This resulted low predictive power but a high precision due to the low number of nominated genes compared to the other V2G datasets. As suggested by referee 1, we have also subjected our data to the ‘activity-by-contact’ (ABC) algorithm and have included this dataset in the comparison as well. Please see Figure 5 in the revised manuscript.

We have elected not to include data from the other studies suggested by the referee for the following reasons: The stimulation paradigm used in the Glinos study is very different from that used in other studies. Also, this study and the study by Calderon did not nominate genes. The studies by Alasoo et al. and Kim-Hellmuth et al. analyzed macrophages, which are not a comparable cell type to CD4+ T cells. The allele-specific eQTL study by Gutierrez-Arcelus et al. included relevant the cell type and activation states, but included a relatively small number of samples (24) and variants (561), and the raw data in dbGAP does not readily allow for re-analysis and harmonization with the other studies. We thank the reviewer for helpful suggestions that have improved the quality of our study.

**Reviewer #3 (Public Review):**
Summary:This paper used RNAseq, ATACseq, and Hi-C to assess gene expression, chromatin accessibility, and chromatin physical associations for native CD4+ T cells as they respond to stimulation through TCR and CD28. With these data in hand, the authors identified 423 GWAS signals to their respective target genes, where most of these were not in the proximal promoter, but rather distal enhancers. The IL-2 gene was used as an example to identify new distal cisregulatory regions required for optimal IL-2 gene transcription. These distal elements interact with the proximal IL2 promoter region. When the distal enhancer contained an autoimmune SNP, it affected IL-2 gene transcription. The authors also identified genetic risk variants that were associated with genes upon activation. Some of these regulate proliferation and cytokine production, but others are novel.Strengths:This paper provides a wealth of data related to gene expression after CD4 T cells are activated through the TCR and CD28. An important strength of this paper is that these data were intensively analyzed to uncover autoimmune disease SNPs in cis-acting regions. Many of these could be assigned to likely target genes even though they often are in distal enhancers. These findings help to provide a better understanding concerning the mechanism by which GWAS risk elements impact gene expression.Another strength of this study was the proof-of-principle studies examining the IL-2 gene. Not only were new cis-acting enhancers discovered, but they were functionally shown to be important in regulating IL-2 expression, including susceptibility to colitis. Their importance was also established with respect to such distal enhancers harboring disease-relevant SNPs, which were shown to affect IL-2 transcription.The data from this study were also mined against past CRISPR screens that identified genes that control aspects of CD4 T cell activation. From these comparisons, novel genes were identified that function during T cell activation.Weaknesses:A weakness of this study is that few individuals were analyzed, i.e., RNAseq and ATACseq (n = 3) and HiC (n = 2). Thus, the authors may have underestimated potentially relevant risk associations by their chromatin capture-based methodology. This might account for the low overlap of their data with the eQTL-based approach or the HIEI truth set.Impact:This study indicates that defining distal chromatin interacting regions helps to identify distal genetic elements, including relevant variants, that contribute to gene activation.

We greatly appreciate the referee’s expert assessment of our work and emphasis on the value of our functional follow-up studies. We have ensured that all sample sizes, effect sizes, p values and FDR statistics are included in the figures and figure legends. We agree that including more donors for the HiC studies would increase the number of implicated variants and genes, however, all the chromatin-based V2G approaches described in our manuscript use relatively small sample sizes, but implicate more variants and genes than the comparable eQTL studies. I.e., the low overlap is not driven by a paucity of GWAS-chromatin-based associations. An alternative explanation for the low overlap between GWAS-chromatin-based approaches and eQTL approaches was recently by Pritchard and colleagues, who reported that GWAS and eQTL studies systematically implicate different types of variants (Mostafavi et al., Nature Genetics 2023). Among other differences, eQTL tend to implicate nearby genes while GWAS variants implicate distant genes, and our results support this contention. We referred to this study in the original version of the manuscript, but have included a more extensive discussion of potential explanations in the revised version. We thank the reviewer for helpful suggestions that have improved the quality of our study.